# Genetic analysis of the septal peptidoglycan synthase FtsWI complex supports a conserved activation mechanism for SEDS-bPBP complexes

Ying Li[1], Han Gong[1], Rui Zhan[1], Shushan Ouyang[1], Kyung-Tae Park[2], Joe Lutkenhaus[2]*, Shishen Du[1]*

**1** Department of Microbiology, Hubei Key Laboratory of Cell Homeostasis, College of Life Sciences, Wuhan University, Wuhan, HB, China, **2** Department of Microbiology, Molecular Genetics and Immunology, University of Kansas Medical Center, Kansas City, KS, United States of America

\* jlutkenh@kumc.edu (JL); ssdu@whu.edu.cn (SD)

**Data Availability Statement:** All relevant data are within the manuscript and its Supporting Information files.

## Abstract

SEDS family peptidoglycan (PG) glycosyltransferases, RodA and FtsW, require their cognate transpeptidases PBP2 and FtsI (class B penicillin binding proteins) to synthesize PG along the cell cylinder and at the septum, respectively. The activities of these SEDS-bPBPs complexes are tightly regulated to ensure proper cell elongation and division. In *Escherichia coli* FtsN switches FtsA and FtsQLB to the active forms that synergize to stimulate FtsWI, but the exact mechanism is not well understood. Previously, we isolated an activation mutation in *ftsW* (M269I) that allows cell division with reduced FtsN function. To try to understand the basis for activation we isolated additional substitutions at this position and found that only the original substitution produced an active mutant whereas drastic changes resulted in an inactive mutant. In another approach we isolated suppressors of an inactive FtsL mutant and obtained FtsW[E289G] and FtsI[K211I] and found they bypassed FtsN. Epistatic analysis of these mutations and others confirmed that the FtsN-triggered activation signal goes from FtsQLB to FtsI to FtsW. Mapping these mutations, as well as others affecting the activity of FtsWI, on the RodA-PBP2 structure revealed they are located at the interaction interface between the extracellular loop 4 (ECL4) of FtsW and the pedestal domain of FtsI (PBP3). This supports a model in which the interaction between the ECL4 of SEDS proteins and the pedestal domain of their cognate bPBPs plays a critical role in the activation mechanism.

## Author summary

Bacterial cell division requires the synthesis of septal peptidoglycan by the widely conserved SEDS-bPBP protein complex FtsWI, but how the complex is activated during cell division is still poorly understood. Previous studies suggested that FtsN initiates a signaling cascade in the periplasm to activate FtsWI. Here we isolated and characterized activated FtsW and FtsI mutants and confirmed that the signaling cascade for FtsW

**Funding:** This study was supported by funding from National Institute of General Medical Sciences (grant GM029764; https://www.nigms.nih.gov/) to JL, and funding from National Natural Science Foundation of China (grant 32070032, http://www.nsfc.gov.cn/) and Wuhan University (https://www.whu.edu.cn/) to SD. The funders had no role in study design, data collection and analysis, decision to publish, or preparation of the manuscript.

**Competing interests:** The authors have declared that no competing interests exist.

activation goes from FtsN to FtsQLB to FtsI and then to FtsW. The residues corresponding to mutations affecting FtsWI activation are clustered to a small region of the interaction interface between the pedestal domain of FtsI and the extracellular loop 4 of FtsW, suggesting that this interaction mediates activation of FtsW. This is strikingly similar to the proposed activation mechanism for the RodA-PBP2 complex, another SEDS-bPBP complex required for cell elongation. Thus, the two homologous SEDS-bPBP complexes are activated similarly by completely unrelated activators that modulate the interaction interface between the SEDS proteins and the bPBPs.

## Introduction

The bacterial divisome is a multi-protein complex that synthesizes septal peptidoglycan during cell division in most walled bacteria [1,2]. For decades, the class A penicillin binding protein PBP1b (contains both glycosyltransferase and transpeptidase activity) in *E. coli* was thought to be the primary septal PG synthase [2]. However, recent studies indicate that FtsW is the essential septal PG glycosyltransferase which functions along with its cognate transpeptidase FtsI (PBP3) [3,4], a class B penicillin binding protein. PBP1b contributes to septal PG synthesis, but it is not essential since cells divide relatively normally in its absence [4–10]. FtsW is a member of the SEDS (shape, elongation, division and sporulation) family and is widely conserved in bacteria having a cell wall [11,12]. It contains 10 transmembrane segments and forms a complex with FtsI to synthesize septal PG [3,13]. Another SEDS-bPBP complex, formed by RodA and PBP2, synthesizes PG during cell elongation in many rod-shaped bacteria [12,14,15]. Although the divisome and elongasome each contain a SEDS-bPBP pair connected to an actin-like protein (FtsA and MreB, respectively), there are many additional components involved in regulation of these two systems (FtsQLBN *vs* MreCD/RodZ). Moreover, these regulators are completely unrelated. Elucidating how these disparate regulators activate these homologous complexes to synthesize PG is critical for understanding cell elongation and division.

An *in vitro* PG polymerization assay showed that PBP2/FtsI is necessary for the PG glycosyltransferase activity of RodA/FtsW [3,15,16], however, the mechanism by which PBP2/FtsI stimulates RodA/FtsW is not fully understood. A structure of the RodA-PBP2 complex from *Thermus thermophilus* revealed that there are two interaction interfaces between RodA and PBP2 (Fig 1A) [16]. One occurs in the plane of the membrane, between the single transmembrane segment of PBP2 and transmembrane segments 8 and 9 of RodA, and is important for complex formation [16]. Another occurs between the periplasmic pedestal domain of PBP2 and extracellular loop 4 (ECL4) of RodA [16]. A negative-stain electron microscopy analysis of the *Tt*RodA-PBP2 complex suggests that this second interaction interface is transient and flexible, with very few specific interactions [16]. Nonetheless, disruption of this interaction abolished PBP2's ability to stimulate RodA *in vitro* [16]. Moreover, some mutations within the ECL4 of RodA or the pedestal domain of PBP2 activate the elongasome/Rod complex *in vivo* [15,17]. Thus, a model emerging from these studies is that the pedestal domain of PBP2 acts as a central allosteric hub to regulate the activity of RodA through ECL4 [16]. However, ECL4 is not completely resolved in the structure of the RodA-PBP2 complex (Fig 1A, dash line) and additional evidence is required to confirm this model as well as to unravel the mechanistic details [16]. Moreover, whether this mechanism applies to FtsWI is not clear since the transmembrane segment of FtsI seems to be sufficient for FtsW activity in the *in vitro* PG polymerization assay [3].

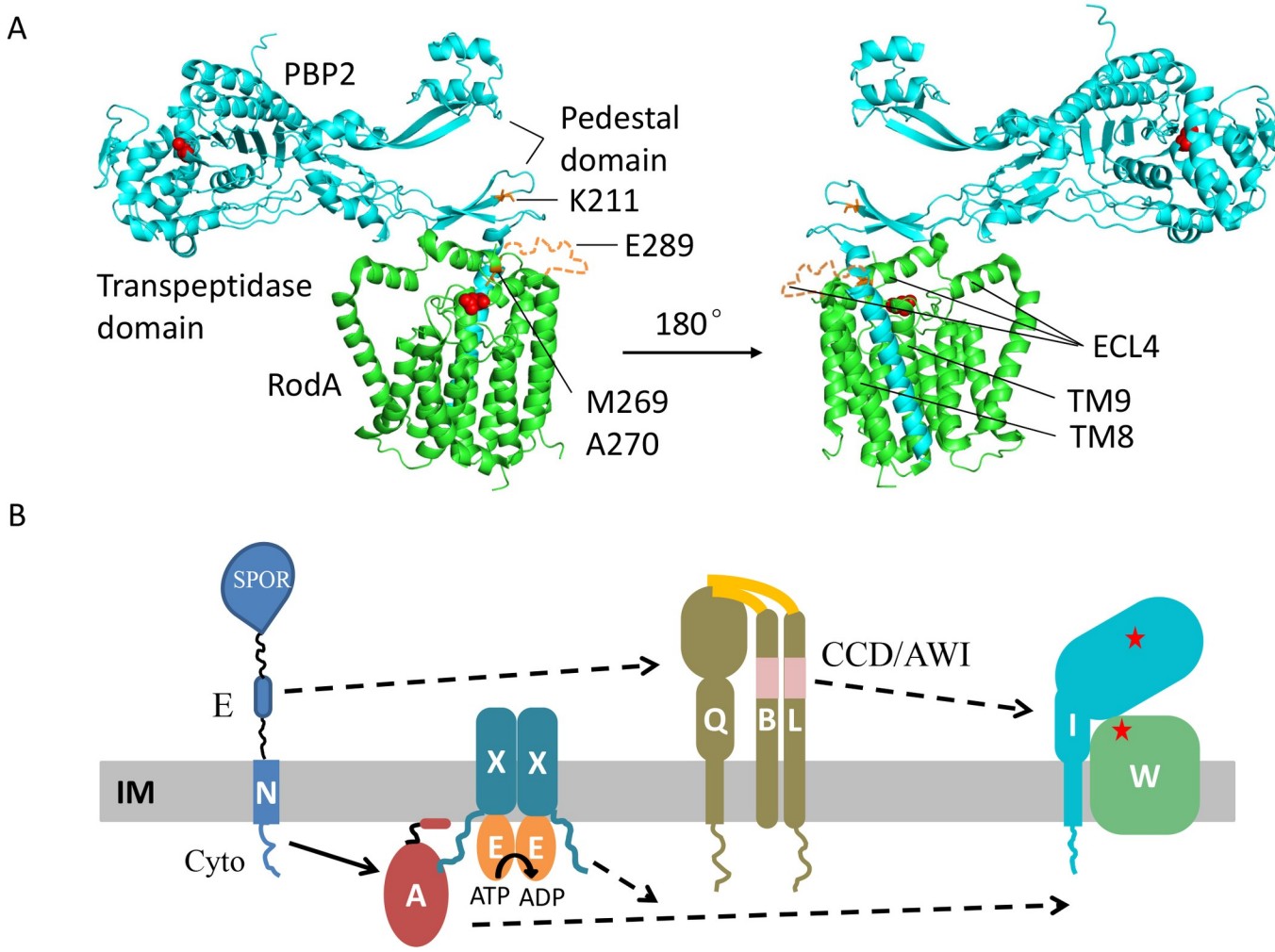

**Fig 1. Structure of a SEDS-bPBP protein complex and a model for FtsN-induced activation of FtsWI.** (A) Structure of *T. thermophilus* RodA-PBP2 (PDB ID#: 6P5L) and the location of the corresponding residues in FtsW and FtsI analyzed in this study. *Tt*RodA is colored green and *Tt*PBP2 is colored cyan, the unstructured region in ECL4 of *Tt*RodA is colored orange. The residues are numbered according to *E. coli* FtsW and the catalytic residues of *Tt*RodA and *Tt*PBP2 are colored red. Residues altered by mutations studied here are indicated and colored orange. (B) The signal transduction pathway leading to the activation of FtsWI. FtsN switches FtsA and FtsQLB to the ON conformations via its cytoplasmic domain (Cyto) and its E domain (E), respectively. In the cytoplasm, the activation signal in FtsA is transmitted to FtsW and this is regulated by the ATPase activity of FtsEX via an interaction between FtsX and FtsA. In the periplasm, the action of the E domain of FtsN likely causes a conformational change of the FtsQLB complex at the CCD domain of FtsB and FtsL, resulting in the AWI domain (**A**ctivation of Fts**WI**) of FtsL becoming available to interact with FtsI. These signals synergize to activate FtsW.

Although FtsI is sufficient for the PG glycosyltranferase activity of FtsW *in vitro* [3], this appears to be a basal activity and another layer of regulation exists *in vivo*. This layer includes FtsA, FtsEX, FtsQLB, and FtsN (Fig 1B) with FtsN arriving last and triggering FtsWI activity [18–25]. FtsN is a bitopic membrane protein with a short N-terminal cytoplasmic domain (FtsN^cyto), a transmembrane segment, followed by an essential domain (FtsN^E) and a SPOR domain in the periplasm [26]. In the current model, FtsN^cyto interacts with FtsA in the cytoplasm and synergizes with the FtsN^E domain in the periplasm to switch FtsA and FtsQLB to the ON states which derepress or activate the enzymatic activity of FtsWI (Fig 1B) [1,18,27].

This regulatory model is based on the isolation of 'activation' (superfission) mutations in *ftsA*, *ftsB* and *ftsL* which lessen or bypass the requirement for FtsN [18], presumably because they mimic FtsN action. The mutations in *ftsB* and *ftsL* cluster in a small region of the coiled

coil domain of FtsB and FtsL termed CCD (**C**onstriction **C**ontrol **D**omain) [18]. A recent study found that *P. aeruginosa* FtsQLB, or just FtsLB, activates the PG polymerization activity of FtsWI, suggesting that FtsQLB is an activator of FtsWI and not a repressor [21]. However, FtsN did not stimulate the *in vitro* system, suggesting the isolated complex is already in an activated state [21]. In support of this model, FtsL mutants were isolated that still form a complex with FtsWI but are defective for cell division *in vivo* and fail to stimulate FtsWI *in vitro* [21]. Another study isolated similar variants of FtsL in *E. coli* and found that these mutations defined a motif in FtsL named AWI (**A**ctivating Fts**WI**) that is critical for activation of septal PG synthesis [22]. Further characterization of these FtsL mutants suggested a model in which the AWI domain of FtsL is likely to be exposed by FtsN action to act on FtsI to stimulate the PG polymerization activity of FtsW [22]. However, biochemical evidence for the interaction is still missing.

FtsEX has a complex role in the regulation of septal PG synthesis. It interacts with FtsA to promote the recruitment of downstream proteins but blocks septal PG synthesis if it cannot hydrolyze ATP [20,23]. Consistent with this, mutations in *ftsA* that disrupt this interaction alleviate the inhibition caused by an FtsEX ATPase mutant, provided that *ftsA* carries another mutation that bypasses the role of FtsEX in divisome assembly [20]. Mutations in the CCD domain of *ftsL* or *ftsB* that 'activate' the divisome in the periplasm (require less FtsN) also confer resistance to the division inhibitory activity of the FtsEX ATPase mutant which blocks the cytoplasmic signal from FtsN [20]. Interestingly, an FtsW mutant (FtsW[M269I]), isolated as a suppressor of the FtsEX ATPase mutant, is less dependent upon FtsN [20]. The mutation is located in the ELC4 of FtsW, potentially mimicking the FtsL-FtsI interaction and switching FtsW to an active conformation. To learn more about the activation mechanism of FtsWI, we further characterized *ftsW[M269I]* and isolated additional active *ftsW* and *ftsI* mutations in this study. Our results confirm that the signaling cascade for FtsW activation follows the order of FtsN>FtsQLB>FtsI>FtsW and supports a model in which FtsI/PBP2 activates FtsW/RodA via an interaction between their pedestal domain and the ECL4 of FtsW/RodA. This model also explains why in some bacteria FtsI is essential even though its transpeptidase activity is dispensable.

## Results

### Overexpression of FtsW[M269I] bypasses FtsN for division

Previously, we showed that FtsW[M269I] reduces the dependency on FtsN [20], however, it was not clear whether it can completely bypass FtsN. To check this, the growth and morphology of cells carrying either wild type *ftsW* or *ftsW[M269I]* on the chromosome were monitored following depletion of FtsN. The FtsN depletion strain contained a plasmid which expresses FtsN constitutively but is temperature sensitive for replication so that at 42°C the plasmid is unable to replicate and FtsN is gradually depleted. As shown in S1A Fig, the strain expressing FtsW[M269I] grew at 42°C whereas the strain expressing FtsW had greater than a 1000-fold plating deficiency. In liquid culture, cells with *ftsW* were filamentous 4 hours after the shift to 42°C. In contrast, cells expressing FtsW[M269I] had a normal cell size at this time, however, by 8 hours they stopped dividing and became filamentous indicating *ftsN* was not bypassed (S1B Fig). Consistent with this, P1 transduction of the *ftsN::kan* allele into a strain carrying *ftsW[M269I]* yielded colonies of varying sizes but the transductants were unable to grow when restreaked (S1C Fig). These results indicate that FtsW[M269I] is more efficient than FtsW for division and that a chromosomal copy of *ftsW[M269I]* almost, but not completely, bypasses FtsN.

Some activating mutations are able to bypass *ftsN* completely when overexpressed. One example is FtsA[E124A], which was originally reported to bypass FtsN [28], however, subsequent work found that it had to be overexpressed [26]. To see if overexpression of FtsW[M269I]

bypassed *ftsN*, we introduced plasmids expressing either FtsW or FtsW^M269I under the control of an IPTG-inducible promoter into W3110 as well as a derivative carrying *ftsW^M269I* on the chromosome, and then tried to knock out *ftsN* by P1 transduction. For a control we used a plasmid that expresses FtsN under an IPTG-inducible promoter. As shown in S2 Fig, *ftsN*::*kan* transductants were readily obtained in the presence or absence of IPTG when the control plasmid expressing *ftsN* was present (basal expression of *ftsN* from the plasmid is sufficient for complementation). As expected, *ftsN*::*kan* could not be transduced into W3110 or the strain with *ftsW^M269I* on the chromosome when they contained a plasmid expressing *ftsW*. However, transductants were readily obtained with both the wild type and the FtsW^M269I mutant strains when *ftsW^M269I* was on the plasmid and induced with 60 μM IPTG (S2 Fig). These transductants grew well when restreaked in the presence of IPTG but poorly in its absence (Fig 2A). In liquid culture, cells became filamentous if FtsW^M269I was not induced, whereas cells overexpressing FtsW^M269I continued to divide, although these cells were longer than wild type cells (Fig 2B). These results demonstrate that overexpression of *ftsW^M269I* can bypass FtsN.

## Mutations at the M269 position can activate or inactivate FtsW

The above results demonstrate that FtsW^M269I is an activated mutant which can bypass FtsN provided it is overexpressed, but how it works is not clear. Alignment of FtsW and its paralog

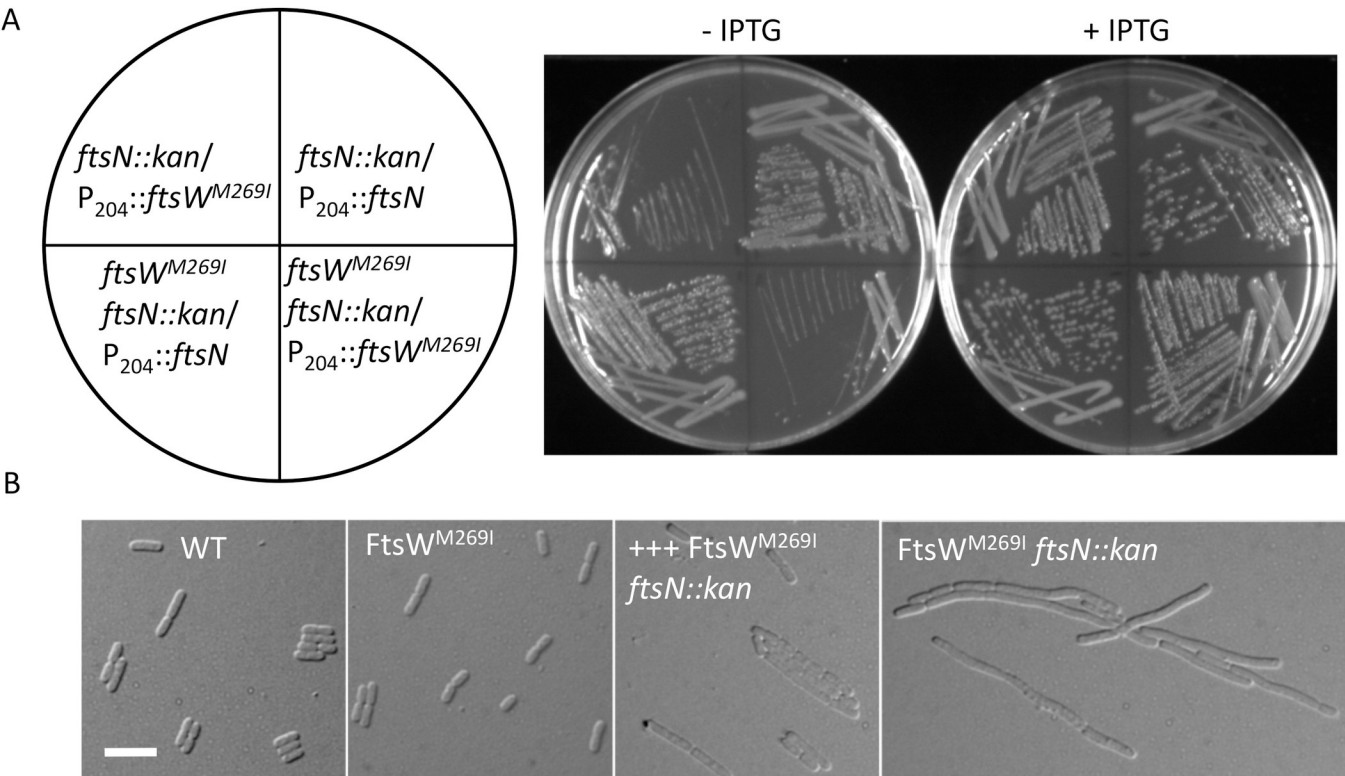

**Fig 2. Overexpression of FtsW^M269I bypasses FtsN.** (A) IPTG-dependent growth of *ftsN*::*kan* transductants from S2 Fig. A single transductant of each strain was restreaked on LB plates with ampicillin, kanamycin, 1 mM sodium citrate and with or without 60 μM IPTG. Plates were incubated at 37°C overnight and photographed. (B) Morphology of WT and FtsW^M269I cells with or without FtsN. Cells from an overnight culture of SD534 [W3110, *ftsN*::*kan ftsW^M269I*/ pSEB429-M269I (pDSW208, P_204::*ftsW^M269I*)] were collected by centrifugation and washed twice with LB to remove IPTG and resuspended in the same volume of LB. The culture was diluted 1:100 and split in half, one with IPTG and one without IPTG and grown at 37°C. The control cultures (W3110 and SD488 [W3110, *leu*::*Tn10 ftsW^M269I*]) were diluted 1:100 in fresh LB medium with antibiotics and grown at 37°C. Samples were taken for photography 2.5 hours after removal of IPTG. The scale bar is 5 μm.

RodA from diverse bacterial species revealed that M269 is not very conserved, but in most cases the position contains a hydrophobic amino acid (S3 Fig). Interestingly, the neighboring residue A270 is highly conserved and an A246T mutation in *Caulobacter crescentus* FtsW (corresponding to A270 of *E. coli* FtsW) produces a hyperactive division phenotype [29,30]. Also, mutation of the corresponding residue in *E. coli* RodA (A234T) bypasses its regulators (RodZ, MreC and MreD) in the elongasome complex [15,17]. In the recently reported *Tt*RodA-PBP2 structure [16], the residues corresponding to M269 and A270 are located in ECL4 of *Tt*RodA and are close to the tip of the pedestal domain of *Tt*PBP2 (Fig 1A), which is believed to function in the allosteric activation of *Tt*RodA. These observations suggest that the M269-A270 region of FtsW and the corresponding region of RodA play an important role in regulating their activities and further characterization of this region may provide insight into the activation mechanism of this protein family.

Since the M269I mutation is a subtle change, we tested whether other changes (M269A, M269V, M269F, M269K and M269E) would result in stronger variants that rescue the growth of an *ftsN* depletion strain without overexpression. If M269I disrupted an interaction to produce an active mutant, a more drastic change should be more disruptive leading to an even more active mutant. As shown in Fig 3A, substitutions that replaced M269 with hydrophobic residues supported the growth of an FtsW depletion strain about as well as the WT. However, they could not rescue the growth of the FtsN depletion strain even when overexpressed, indicating they were not activated mutants (Fig 3B). Removing the side chain of M269 (M269A)

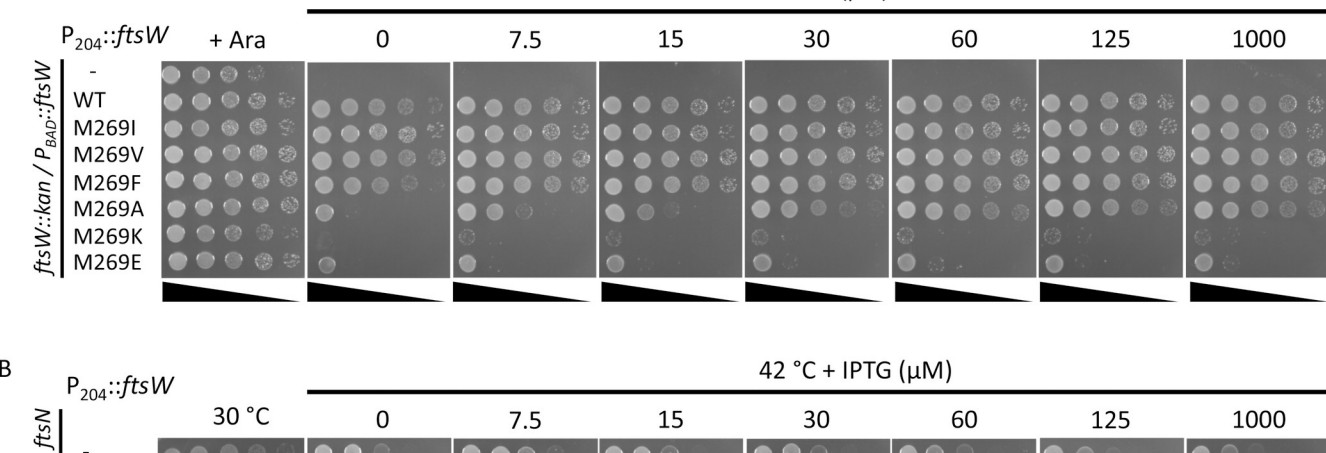

**Fig 3. FtsW$^{M269I}$ is a unique substitution that suppresses an FtsN depletion strain.** (A) Complementation test of FtsW mutants. Plasmid pDSW208, pSEB429 or its derivatives carrying *ftsW* mutations were transformed into strain SD237 [W3110, *leu::Tn10, ftsW::kan*/pDSW406 (pBAD33, P$_{BAD}$::*ftsW*)] on LB plates with ampicillin, kanamycin, chloramphenicol and 0.2% arabinose. The next day, a single transformant of each resulting strain was resuspended in 1 ml of LB medium, serially diluted. 3 μl of each dilution was spot on LB plates with antibiotics, with or without 0.2% arabinose and increasing concentrations of IPTG. Plates were incubated at 37°C overnight and photographed. (B) Spot test of the ability of FtsW mutants to rescue the growth of an FtsN depletion strain. Colonies of strain SD264 [W3110, *ftsN::kan*/pBL154 (pSC101$^{ts}$, P$_{syn}$::*ftsN*, *spc*)] harboring pDSW208 or pSEB429 expressing FtsW variants were resuspended in 1 ml of LB medium and serially diluted. 3 μl of each dilution was spotted on LB plates with antibiotics with or without IPTG. Plates were incubated at 30°C for 24 hours or at 42°C overnight and photographed.

resulted in a partially functional protein that had to be overexpressed to complement whereas changing it to charged amino acids (M269K or M269E) resulted in non-functional proteins (Fig 3A). Not surprisingly, these mutants were also unable to rescue the growth of the FtsN depletion strain. These results suggest that a bulky hydrophobic residue at 269 is very important for the normal function of FtsW and M269I is a unique substitution that leads to an active FtsW. Also, introducing a charged amino acid results in an inactive protein. We also tested the A270T mutation (corresponding to the $Cc$FtsW$^{A246T}$ and $Ec$RodA$^{A234T}$) to see if it was an active mutant. FtsW$^{A270T}$, however, had a slightly reduced function since a higher inducer concentration was required for complementation (Fig 4A). Western blot of GFP fusion of FtsW$^{M269K}$ and FtsW$^{A270T}$ using anti-GFP antibody showed that they were expressed at the same level as wild type FtsW (Fig 4B), indicating that it was unlikely because they were unstable.

## Inactive FtsW mutants localize to division site and recruit downstream proteins

Since FtsW$^{M269K}$ and FtsW$^{A270T}$ did not complement, we tested whether they localized to the Z ring using a depletion strain in which FtsW was supplied from a plasmid under the control of an arabinose inducible promoter and FtsW-GFP fusions were expressed from an IPTG-inducible promoter on a compatible plasmid. As shown in Fig 5A, both FtsW-GFP and FtsW$^{A270T}$-GFP localized to the division site and were able to support division in the absence of arabinose. Note that the level of FtsW$^{A270T}$-GFP expressed from this plasmid is sufficient to overcome the mutant's defect. Although FtsW$^{M269K}$-GFP localized to potential division sites, it did not support division. These results indicate that FtsW$^{M269K}$, and probably FtsW$^{A270T}$, do not have a problem in localizing to the Z ring.

Previous studies have shown that recruitment of FtsI is due to its TM domain and flanking residues interacting with FtsW [31,32]. Since M269 and A270 are in close proximity to the interaction interface between the TM domain of FtsW and FtsI, we tested whether FtsW$^{M269K}$ and FtsW$^{A270T}$ affected the recruitment of FtsI. As expected, GFP-FtsI no longer localized to potential division sites in filaments following FtsW depletion, and adding back FtsW or FtsW$^{A270T}$ restored cell division and GFP-FtsI localization (Fig 5B). Although FtsW$^{M269K}$ was unable to support division, it still recruited GFP-FtsI to presumptive division sites in the filaments, indicating that the divisome was assembled. Furthermore, bacterial two hybrid analysis showed that M269K and A270T did not affect the interaction between FtsW and FtsI or

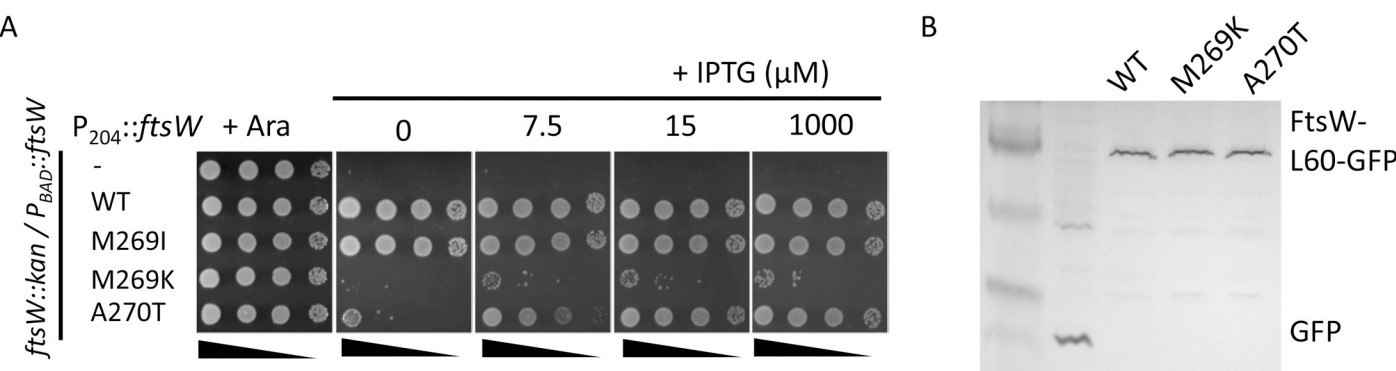

**Fig 4. Analysis of FtsW mutants.** (A) Complementation test of FtsW mutants. Plasmid pDSW208, pSEB429 or its derivatives carrying $ftsW$ mutations were transformed into strain SD237 [W3110, $leu$::Tn10, $ftsW$::$kan$/pDSW406 (pBAD33, P$_{BAD}$::$ftsW$) on LB plates with ampicillin, kanamycin, chloramphenicol and 0.2% arabinose. The spot test was performed as Fig 3A. (B) Western blot to test the stability of FtsW mutants. Details about the blot is described in Materials and Methods.

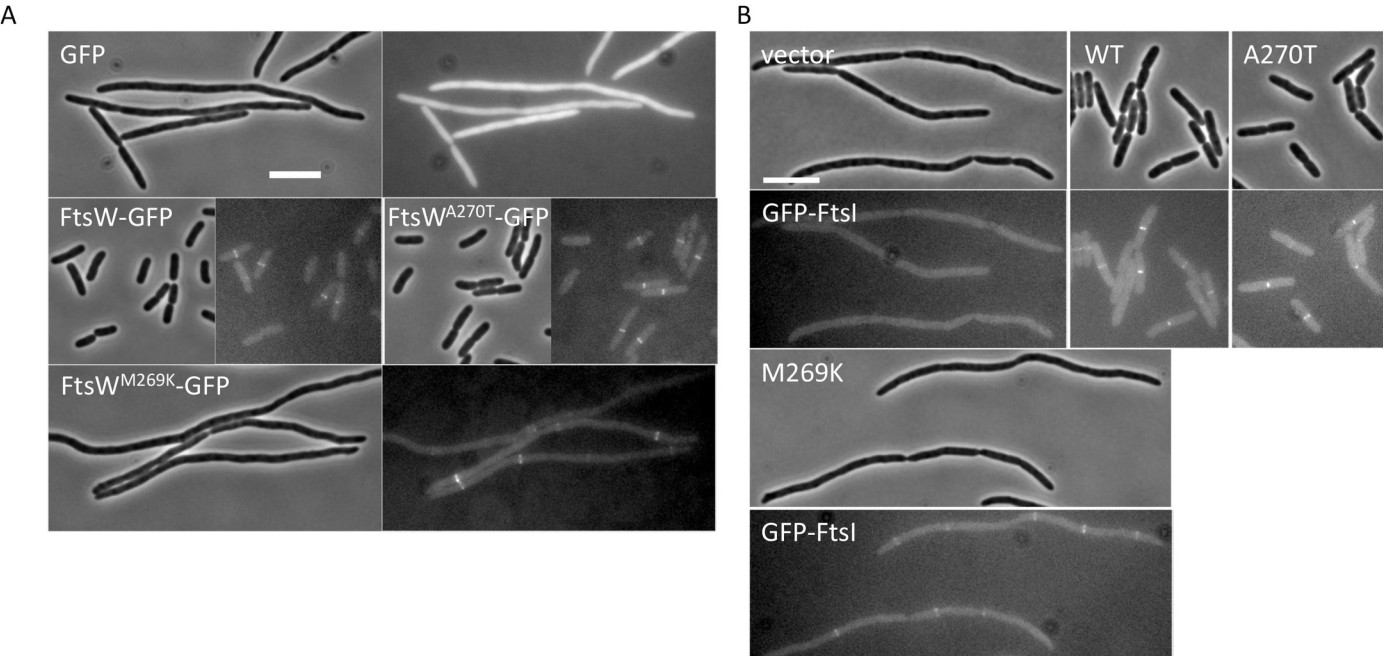

**Fig 5. FtsW^M269K and FtsW^A270T localize to the division site and recruit FtsI.** (A) Localization of FtsW mutants. Localization of the FtsW mutants was assessed in the FtsW depletion strain SD237 [W3110, *leu*::Tn*10 ftsW*::*kan/pDSW406 (*P_BAD_::*ftsW)]* using plasmid pSD347 (P_206_::*ftsW-l60-gfp*) and derivatives carrying *ftsW* mutations. After removal of arabinose, IPTG was added to the culture to induce the fusion protein and 4 hours post induction cells were immobilized on 2% agarose pad for photographing. (B) Localization of FtsI. Overnight cultures of SD288 [W3110, *ftsW*::*kan gfp-ftsI*/pSD257 (pSC101^ts, *ftsW*)] carrying plasmid pBAD33 or pDSW406 (pBAD33, P_BAD_::*ftsW*) or its derivatives grown at 30°C were diluted 1:100 in fresh LB medium with antibiotics and grown at 30°C for 2 h. The cultures were then diluted 1:10 in fresh LB medium with antibiotics, 0.2% arabinose and 10 μM IPTG and grown at 37°C for 4 hours. Cells were immobilized on 2% agarose pads for photography. The scale bar is 5 μm.

between FtsW and FtsQLB (S4 Fig). The active mutant M269I also displayed a similar interaction with these divisome proteins. Thus, the mutations (active or inactive) did not detectably affect the interaction between FtsW and other divisome proteins, indicating that they likely affect the activation of FtsW.

## Isolation of FtsW^E289G and FtsI^K211I as hyperactive division mutants that bypass FtsN

In a recent study, we isolated dominant negative FtsL mutants that block cell division [22]. These mutants support assembly of the divisome but are unable to divide because they are likely unable to interact with FtsI in response to FtsN [22]. Such mutants are suppressed by the hyperactive *ftsW^M269I* mutation which no longer requires the activation signal from FtsN [22]. To isolate additional active *ftsW* and *ftsI* mutations, we screened FtsW and FtsI mutant libraries for suppressors of one of these FtsL mutants (FtsL^E87K). We isolated one mutation in ECL4 of FtsW (E289G) and one in the pedestal domain of FtsI (K211I) (Fig 1A). As previously shown and in Figs 6 and S5, FtsL^E87K and another inactive mutant FtsL^L86F were unable to complement an FtsL depletion strain when expressed from an arabinose-inducible promoter on a plasmid. However, expression of FtsW^E289G or FtsI^K211I from an IPTG-inducible promoter on a compatible plasmid rescued the growth of the strain when FtsL^E87K or FtsL^L86F were expressed, indicating that they are suppressed.

To prove that FtsW^E289G and FtsI^K211I were active division mutants, we tested if they suppressed other mutations that blocked septal PG synthesis. First, we tested the ATPase defective

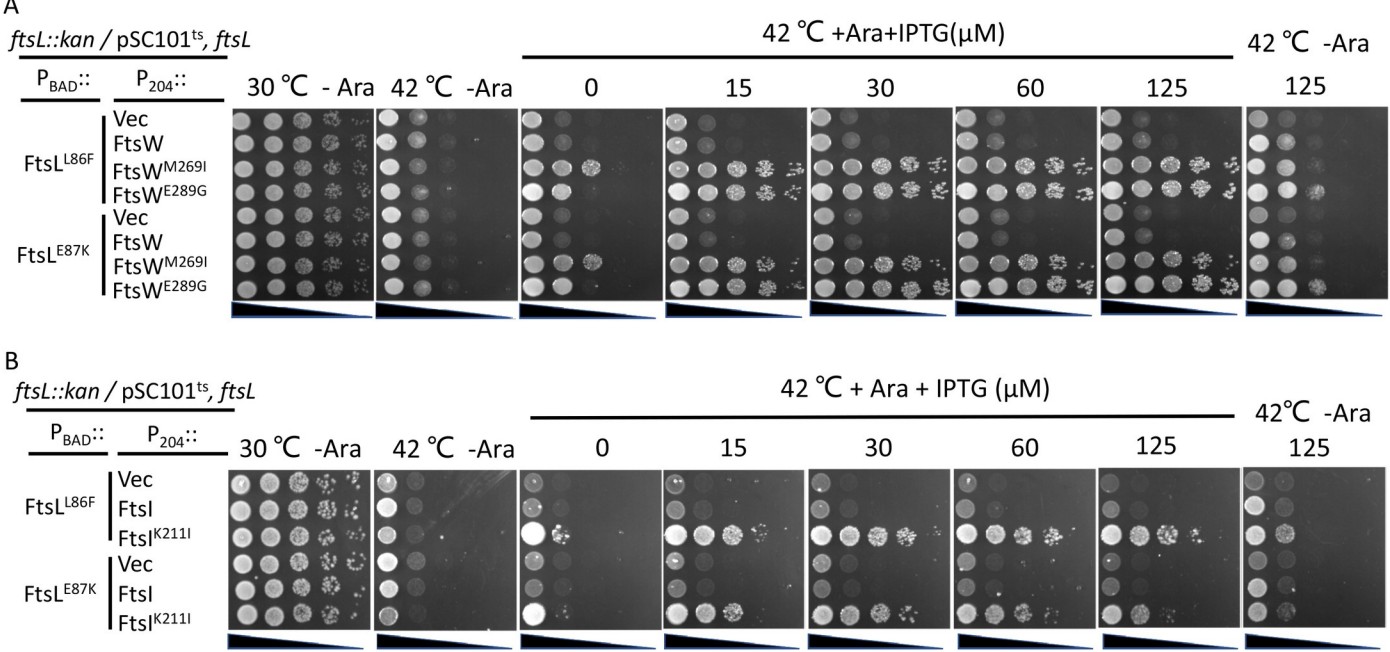

**Fig 6. FtsI$^{K211I}$ and FtsW$^{E289G}$ rescue inactive FtsL mutants.** Plasmids pDSW208, pSEB429 (pDSW208, P$_{204}$::$ftsW$), pLY91 (pDSW208, P$_{204}$::$ftsI$) or their derivatives carrying different $ftsW$ or $ftsI$ alleles were transformed into strain SD399 (W3110, $ftsL$::$kan$ /pSD256) harboring plasmid pSD296-L86F (pBAD33, P$_{BAD}$::$ftsL^{L86F}$) or pSD296-E87K (pBAD33, P$_{BAD}$::$ftsL^{E87K}$) and transformants selected on LB plates with antibiotics and glucose. The next day, a single colony of each resulting strain was subject to spot tests. Plates were incubated at 30˚C for 24 hours or at 42˚C overnight and photographed.

mutant of FtsEX, FtsE$^{D162N}$X, which was previously shown to block septal PG synthesis and to be suppressed by FtsW$^{M269I}$ [20,23,33]. As shown in S6A Fig, overexpression of FtsEX$^{D162N}$ blocked the growth of the wild type strain but not a strain carrying the $ftsW^{E289G}$ mutation. Notably, FtsW$^{E289G}$'s resistance to FtsE$^{D162N}$X was even greater than FtsW$^{M269I}$, suggesting that FtsW$^{E289G}$ is a stronger activation mutant than FtsW$^{M269I}$. A strain carrying the $ftsI^{K211I}$ mutation also displayed resistance to FtsE$^{D162N}$X (S6B Fig), consistent with it being an activation mutation.

Next, we tested if FtsW$^{E289G}$ and FtsI$^{K211I}$ could suppress the depletion of FtsN. As shown in S7 Fig, overexpression of FtsW$^{E289G}$ or FtsI$^{K211I}$ in an FtsN depletion strain fully rescued the growth defect, indicating that they suppressed the depletion of FtsN. To further test the strength of FtsW$^{E289G}$ and FtsI$^{K211I}$, we checked whether they could bypass $ftsN$ by transducing $ftsN$::$kan$ into strains carrying these mutations. FtsI$^{K211I}$ was able to bypass $ftsN$ on LB with 1% NaCl at 42˚C (but not at 37˚C) when it was overexpressed but not at the chromosomal level (S8 Fig). In liquid culture, $ftsI^{K211I}$ $ftsN$::$kan$ cells grew poorly and were filamentous even when $ftsI^{K211I}$ was overexpressed (Fig 7A). On the other hand, a chromosomal copy of $ftsW^{E289G}$ was sufficient to bypass FtsN even on LB with 0.5% NaCl and these transductants grew well upon restreaking (S8B Fig). Inspection of $ftsW^{E289G}$ $ftsN$::$kan$ cells revealed that they are only slightly longer than the wild type cells (Fig 7B), indicating that FtsN was largely bypassed.

We also tested additional substitutions of E289 of FtsW (E289Q, E289L and E289R) and K211 of FtsI (K211A, K211T, K211F, K211E) to see if they could rescue the growth of the FtsN depletion strain. Unlike substitutions at the M269 position of FtsW, which could be beneficial or detrimental for cell division, substitutions at the E289 position did not significantly affect the function of FtsW. Likewise, substitutions at K211 did not affect FtsI function. As shown in

**Fig 7. FtsI$^{K211I}$ and FtsW$^{E289G}$ bypass FtsN.** (A) Morphology of WT and FtsI$^{K211I}$ cells with or without FtsN. Overnight cultures of W3110, LYA8 (W3110, *leu::Tn10 ftsI$^{K211I}$*) and LYA9/pLY105 [W3110, *leu::Tn10 ftsN::kan ftsI$^{K211I}$/pLY105 (P$_{204}$::ftsI$^{K211I}$)*] were diluted 1:100 in fresh LB medium with antibiotics and grown at 42˚C. 60 μM IPTG was added to the culture of LYA9/pLY105. 2 hours later, samples were taken for photography. The scale bar is 5 μm. (B) Morphology of WT and FtsW$^{E289G}$ cells with or without FtsN. Overnight cultures of W3110, SD488, SD264 (W3110, *leu::Tn10 ftsN::kan*/pBL154 (pSC101$^{ts}$, P$_{syn}$::*ftsN*)) and SD530 (W3110, *leu::Tn10 ftsN::kan ftsW$^{E289G}$*) were diluted 1:100 in fresh LB medium with antibiotics and grown at 37˚C. Samples were taken for photography 2 hours later. The scale bar is 5 μm.

S9 and S10 Figs, all tested FtsW and FtsI mutants were functional in a complementation test, but only FtsW$^{E289G}$ and FtsI$^{K211I}$ could rescue the FtsN depleted strain. Thus, similar to *ftsW$^{M269I}$*, only very specific mutations (*ftsW$^{E289G}$* and *ftsI$^{K211I}$*) gain the ability to bypass FtsN.

Lastly, we tested if the *ftsW$^{E289G}$* and *ftsI$^{K211I}$* mutations affected the interaction of FtsW or FtsI with other divisome proteins using the bacterial two hybrid assay. As shown in S4B Fig, *ftsW$^{E289G}$* and *ftsI$^{K211I}$* did not detectably affect the interaction between FtsW or FtsI and other division proteins (FtsL, FtsB, FtsQ and FtsI or FtsW). Thus, these two mutations are unlikely to increase the activity of FtsW *in vivo* by enhancing their interaction with other divisome proteins.

## Epistatic analysis of the division mutations confirms the signaling cascade for FtsWI activation

Our recent study on the role of FtsL in *E. coli* cell division suggests that the periplasmic signaling cascade for FtsWI activation follows the order FtsN>FtsQLB>FtsI>FtsW [22]. In this study, and consistent with this model, we found several active mutants of FtsI and FtsW that suppressed inactive FtsL mutants and bypassed FtsN [22]. If the model is correct, active *ftsW* mutations should also suppress *ftsI* mutations deficient in activation as they would bypass this step. Several mutations in the pedestal domain of FtsI (G57D, S61F, L62P and R210C) were reported to affect the localization of FtsN [34], but our recent study suggested that these

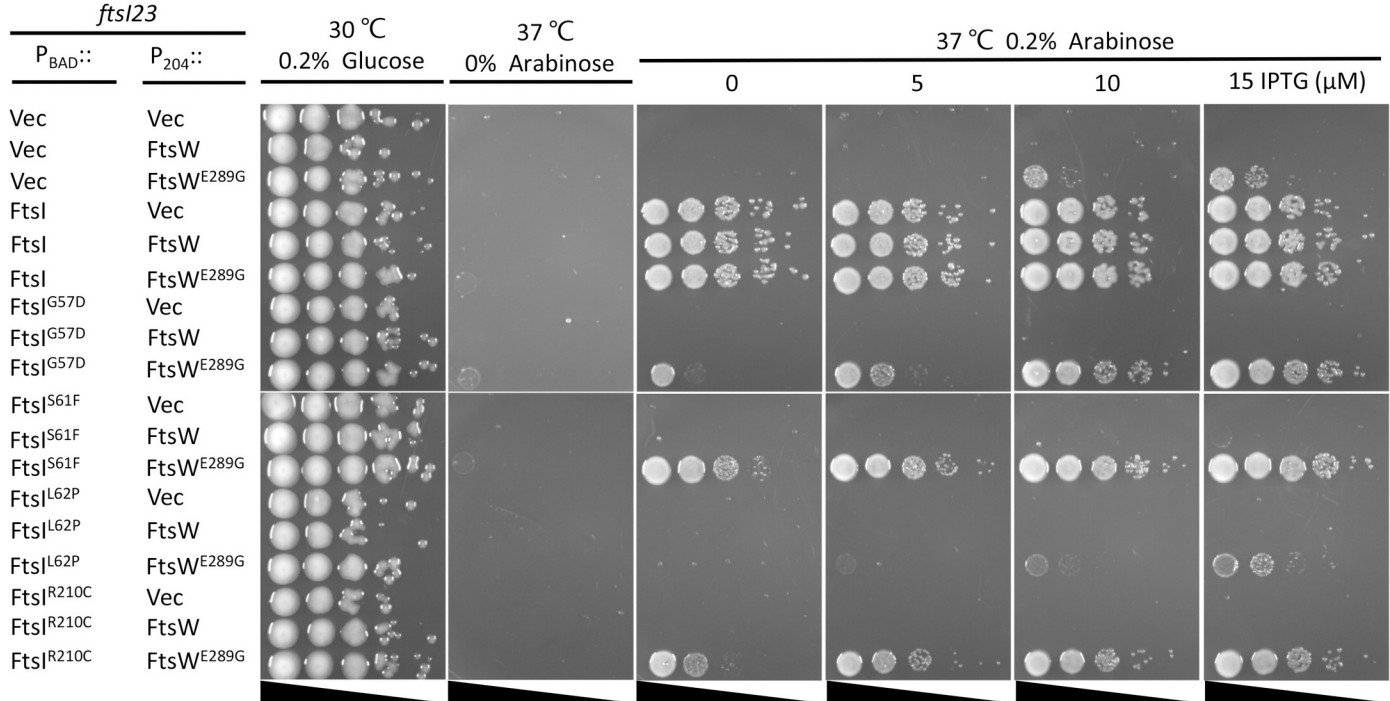

**Fig 8. Overexpression of FtsW<sup>E289G</sup> suppresses FtsI mutations defective in activation of septal PG synthesis.** Plasmid pDSW208, pSEB429 (pDSW208, P<sub>204</sub>:: *ftsW*) or its derivatives carrying different *ftsW* alleles were co-transformed with plasmid pBAD33 carrying different *ftsI* alleles into strain MCI23 *ΔrecA* (MC4100, *ftsI23 recA::spec*). Transformants were selected on LB plates with antibiotics and glucose at 30˚C. The next day, a single colony of each resulting strain was subjected to a spot test.

mutations likely affect the ability of FtsI to activate FtsW [22]. Thus, we tested if FtsW<sup>E289G</sup> could suppress these *ftsI* mutations. As shown in Fig 8, these FtsI mutants were unable to complement a strain carrying the temperature-sensitive *ftsI23* mutation at 37˚C. However, growth was restored for three of the four mutants (G57D, S61F and R210C) if FtsW<sup>E289G</sup> was present. A failure to rescue the fourth mutant (L62P) may be due to the proline substitution which often affects protein structure substantially. Thus, FtsW<sup>E289G</sup> was dominant over three of these inactive FtsI mutations.

Based on our analysis above, FtsW<sup>M269K</sup> and FtsW<sup>A270T</sup> are likely defective in the activation step with FtsW<sup>M269K</sup> being more severely affected than FtsW<sup>A270T</sup>. If the signaling cascade model is correct, activation mutations in upstream proteins in the cascade should not suppress FtsW<sup>M269K</sup> or FtsW<sup>A270T</sup>. We tested two active mutations in the periplasmic cascade, one in FtsB (E56A) and one in FtsI (K211I). As expected, FtsW<sup>M269K</sup> was not suppressed by an active *ftsB<sup>E56A</sup>* mutation or when FtsI<sup>K211I</sup> was overexpressed (Fig 9). However, the growth of the strain containing FtsW<sup>A270T</sup> was restored in the presence of FtsB<sup>E56A</sup> or when FtsI<sup>K211I</sup> was overexpressed. We also tested if overexpression of FtsN or the presence of the *ftsA<sup>R286W</sup>* mutation, both of which enhance activation of FtsWI, could suppress FtsW<sup>M269K</sup> or FtsW<sup>A270T</sup>. As shown in S11 Fig, the growth defect of FtsW<sup>A270T</sup> but not FtsW<sup>M269K</sup> was suppressed. Thus, the weakly defective mutant FtsW<sup>A270T</sup> still responds to activation by FtsN or by activation mutations in the upstream proteins in the signaling cascade, but the severely defective mutant FtsW<sup>M269K</sup> is unable to respond. Altogether, these results support that the signaling cascade flows from FtsN>FtsQLB>FtsI>FtsW and confirm that mutations in downstream proteins are dominant over mutations in upstream proteins.

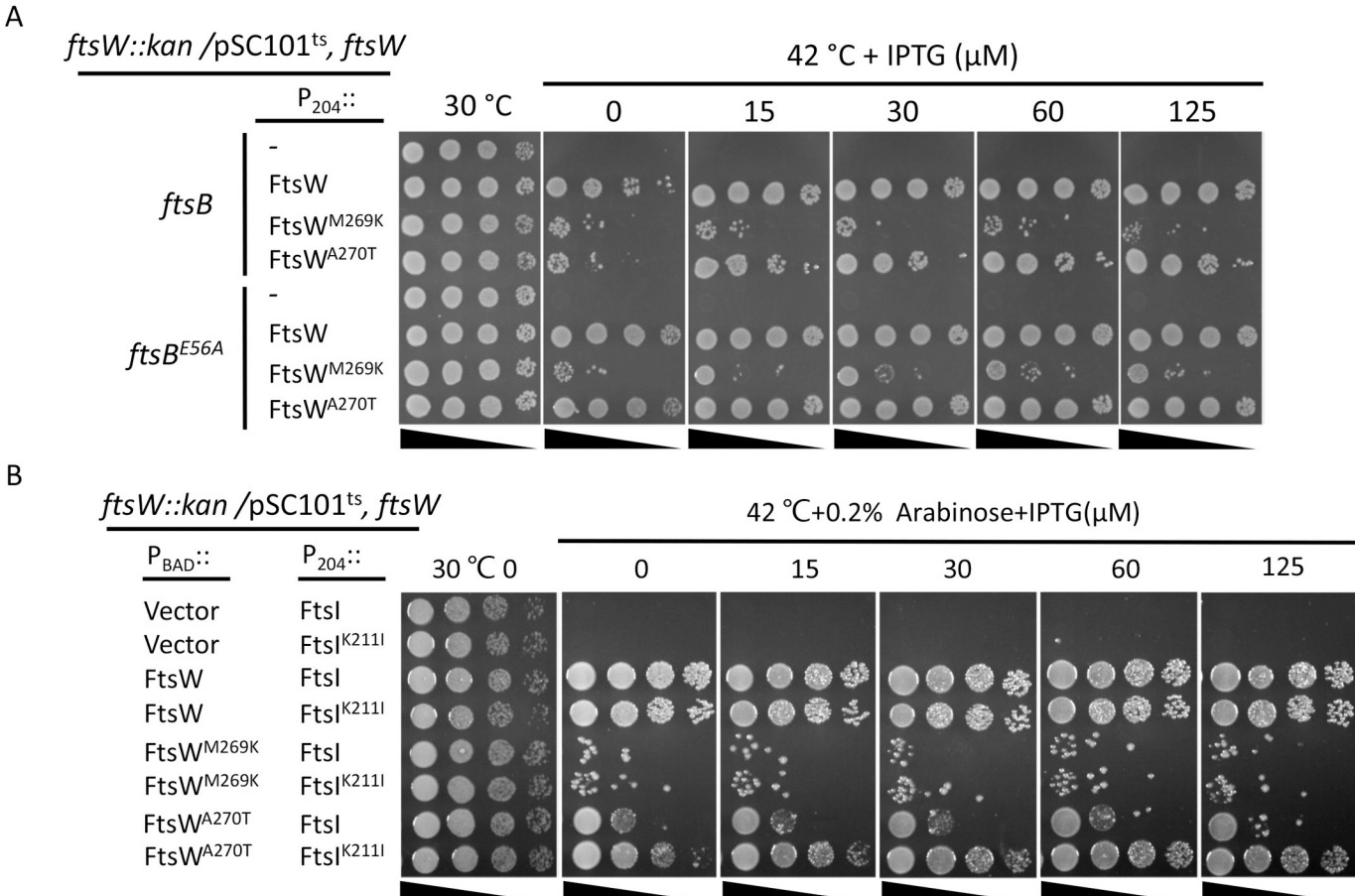

**Fig 9. FtsB$^{E56A}$ and FtsI$^{K211I}$ suppress FtsW$^{A270T}$ but not FtsW$^{M269K}$.** (A) Spot test to assess the ability of FtsB$^{E56A}$ to suppress FtsW$^{A270T}$ and FtsW$^{M269K}$. Plasmid pDSW208, pSEB429 or its derivatives carrying different *ftsW* alleles were transformed into strain SD367 [TB28, *ftsW::kan*/pSD257(pSC101$^{ts}$, *ftsW*)] or SD368 [TB28, *ftsB$^{E56A}$*, *ftsW::kan*/pSD257(pSC101$^{ts}$, *ftsW*)] on LB plates with ampicillin, kanamycin and spectinomycin at 30˚C. The spot test was performed as in Fig 5. (B) Spot test of the ability of FtsI$^{K211I}$ to suppress FtsW$^{A270T}$ and FtsW$^{M269K}$. The test was done in strain SD292 [W3110, *ftsW::kan recA56 slrD::Tn10*/pSD257 (pSC101$^{ts}$, *ftsW*)] containing plasmid pLY91 (P$_{204}$::*ftsI*) or pLY105 (P$_{204}$::*ftsI$^{K211I}$*). SD292 was transformed with plasmid pBAD33 carrying different *ftsW* alleles and transformants selected at 30˚C. Test was done as (A) on LB plates with or without arabinose and IPTG at 30˚C and 42˚C.

## *ftsW$^{E289G}$* and *ftsI$^{K211I}$* are intragenic suppressors of inactive FtsW and FtsI mutants, respectively

In addition to the epistatic analysis of the active and inactive mutations between different division proteins, we also tested the epistatic interaction between the active and inactive mutations of FtsI and FtsW. As shown in Fig 10A, the inactive FtsI mutants (G57D, S61F, L62P and R210C) expressed from an IPTG-inducible promoter on a plasmid were unable to complement the *ftsI23$^{ts}$* strain at 37˚C. However, adding the K211I mutations to these alleles, except for FtsI$^{L62P}$, rescued the growth of the *ftsI23$^{ts}$* strain, indicating that K211I is an intragenic suppressor of these inactivating *ftsI* mutations. However, it is noticeable that a basal level of expression of FtsI or FtsI$^{K211I}$ from the plasmid was sufficient to complement the *ftsI23$^{ts}$* strain, whereas a higher concentration of IPTG was required for complementation with the double mutants, indicating that the inactive mutations reduced the efficiency of FtsI$^{K211I}$. Similarly, the *ftsW$^{E289G}$* mutation suppressed the *ftsW$^{M269K}$* and *ftsW$^{A270T}$* mutations. As shown in Fig 10B, the FtsW depletion strain was unable to grow on plates without arabinose, however, growth was restored by the presence of a plasmid that expressed either wild type FtsW or

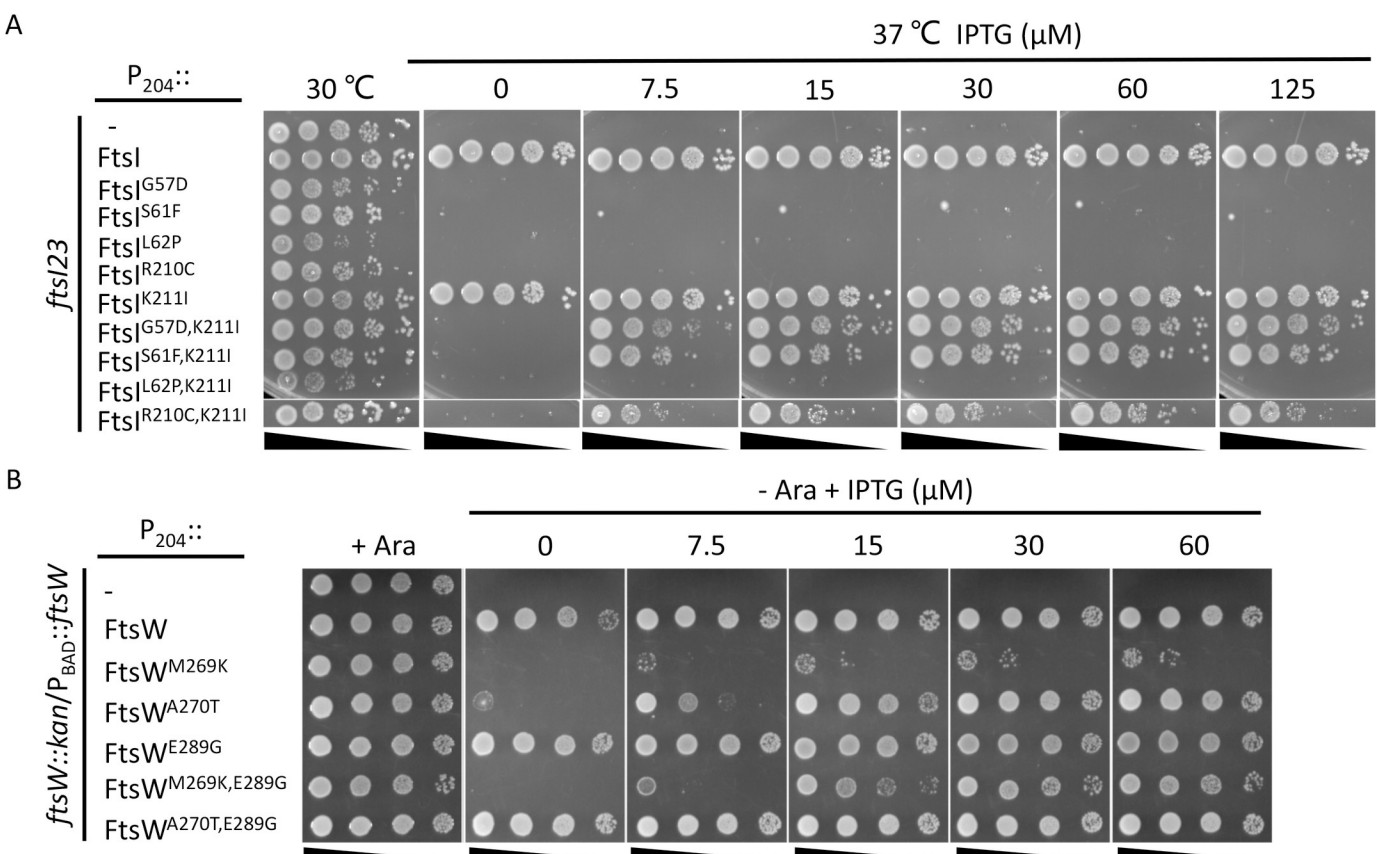

**Fig 10. *ftsI^{K211I}* and *ftsW^{E289G}* are intragenic suppressors of inactive FtsI and FtsW mutants respectively.** (A) *ftsI^{K211I}* suppresses inactive FtsI mutations. To test the ability of various FtsI mutants to be rescued by K211I, plasmid pDSW208, pLY91 (pDSW208, P_{204}::*ftsI*) or its derivatives carrying different alleles of *ftsI* were transformed into strain MCI23ΔrecA (W3110, *recA::spec*, *ftsI23*) at 30°C. Complementation test was done at 42°C (to inactivae FtsI23) and in the presence of IPTG to induce the *ftsI* alleles contained on the plasmids. (B) *ftsW^{E289G}* suppresses inactive FtsW mutants. To test the ability of E289G to rescue deficient FtsW alleles, plasmid pDSW208, pSEB429 or its derivatives carrying different alleles of *ftsW* were transformed into strain SD237 (W3110, *leu::Tn10*, *ftsW::kan*/ pDSW406 [pBAD33, P_{BAD}::*ftsW*]) on LB plates with ampicillin, kanamycin, chloramphenicol and 0.2% arabinose at 37°C. The complementation test was done by removing arabinose (to deplete WT FtsW) and adding IPTG to induce the *ftsW* alleles on the plasmids.

FtsW^{E289G} from an IPTG-inducible promoter (even in the absence of IPTG, indicating that basal expression is sufficient). With FtsW^{A270T} growth was restored at 15 μM IPTG consistent with it being somewhat deficient. However, adding E289G to the deficient alleles (FtsW^{M269K/E289G} and FtsW^{A270T/E289G}) resulted in the complementation of the depletion strain. FtsW^{A270T/E289G} did not require IPTG for complementation, whereas FtsW^{M269K/E289G} needed 15 μM IPTG for optimal growth. Thus, E289G rescued both M269K and A270T, although M269K reduced the activity of E289G.

## Mutations affecting activation of SEDS-bPBP complexes cluster to the interaction interface between the ECL4 of SEDS proteins and the pedestal domain of bPBPs

The residues in FtsW and FtsI that correspond to the mutations analyzed here are either located in the ECL4 of FtsW or in the pedestal domain of FtsI. The corresponding domains have been proposed to play a critical role in activation of the RodA-PBP2 complex [16]. This prompted us to map all known mutations that result in activation or inactivation of the two SEDS-bPBP complexes to the *Tt*RodA-PBP2 structure. These include those in *E. coli* FtsW

**Table 1. Active (orange) and inactive (magenta) FtsW/RodA and FtsI/PBP2 mutations.**

| | EcFtsW | CcFtsW | EcRodA | TtRodA |
|---|---|---|---|---|
| Corresponding residues of mutations in FtsW/RodA | T52 | A31K | I23 | L18 |
| | L164 | F145L | I128 | L122 |
| | V201 | T180A | S163 | A156 |
| | M269I/K/E | I245 | I233 | I226 |
| | A270T | A246T | A234T | A227 |
| | V285 | V261 | T249P | S242 |
| | E289G | R264 | E254 | G247 |
| Corresponding residues of mutations in FtsI/PBP2 | EcFtsI | CcFtsI | EcPBP2 | TtPBP2 |
| | N18 | I45V | A15 | NA |
| | V54 | F71 | Q51L | A35 |
| | K55 | S72 | T52N | L36 |
| | G57D | A74 | S54 | S38 |
| | S61F/P | G78 | R58 | Y42 |
| | L62P | G79 | I59 | L43E |
| | V64 | A81 | L61R | T45 |
| | I208 | NA | E226 | A186E |
| | R210C | NA | E228 | E188 |
| | K211I | NA | V229 | V189 |

NA: Not available (CcFtsI lacks the region containing the corresponding residues in the alignment)

(M269I/K/E, A270T and E289G) and FtsI (G57D, S61F, L62P and R210C, K211I), those in *C. crescentus* FtsW (A31K, F145L, T180A and A246T), those in *E. coli* RodA (A234T and T249P) and PBP2 (Q51L, T52N and L61R), and those in *T. thermophilus* PBP2 (L43E and A186E) (Table 1). The residues altered by most of these mutations, regardless of whether they are active or inactive, are located in a small region of the interaction interface between the ECL4 of *Tt*RodA and the pedestal domain of *Tt*PBP2 or its proximal region (Fig 11). *Ec*FtsW[E289G] and *Ec*RodA[T249P] lie within a small unstructured region of ECL4 which is also near the interaction interface. The location of these mutations on the *Tt*RodA-PBP2 structure support a model in which the pedestal domain of bPBPs is involved in activation of their cognate SEDS proteins via ECL4 and suggest that these residues are critical for the interaction. Notably, some of the same substitutions produced opposite phenotypes in different organisms, for example, *Ec*FtsW[A270T] is partially defective, but *Cc*FtsW[A246T] and *Ec*RodA[A234T] are hyperactive. Also, many mutations are right next to each other but produce opposite phenotypes, for example, FtsI[R210C] is inactive for division, but FtsI[K211I] is more potent for cell division. This suggests that local subtle re-arrangements of these domains lead to the activation or inactivation of the SEDS-bPBP complex. It is also worthwhile to mention that previous evolutionary sequence covariation analyses revealed strong evolutionary coupling between residues within the ECL4 of FtsW and the pedestal domain of FtsI [35–37]. Since mutations that alter these residues do not affect the interaction they must affect the activation.

## Discussion

A key step in bacterial cell division is the activation of septal PG synthesis at the Z ring. In *E. coli* and many gram-negative bacteria, this is triggered by FtsN, the last essential division protein recruited to the Z ring [24–26]. Although FtsN binds to and stimulates the activity the class A penicillin binding protein PBP1b [5,6,9,10], which contributes to septal PG synthesis

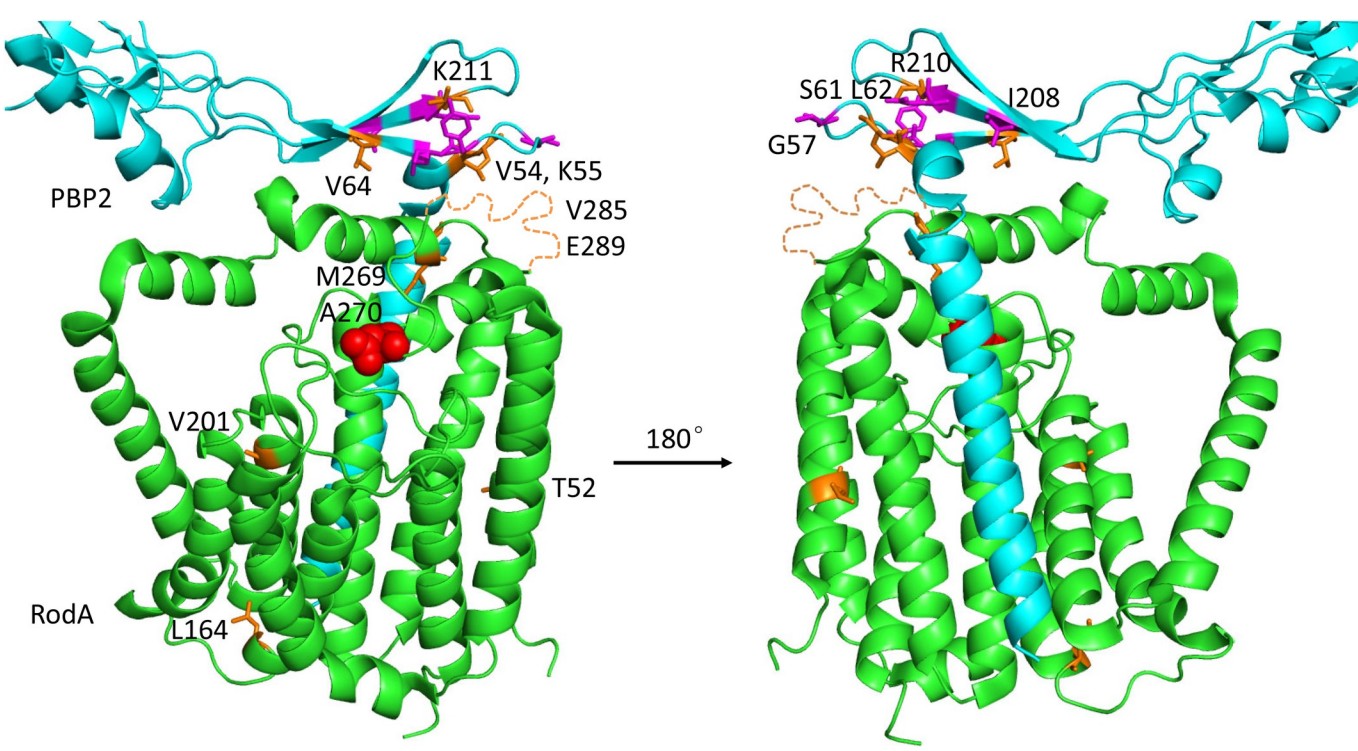

**Fig 11. Location of residues on the *Tt*RodA-PBP2 structure that correspond to mutations that affect the activity of SEDS-bPBP protein complexes.**
*Tt*RodA is colored green, whereas *Tt*PBP2 is colored cyan (only a part of the pedestal domain is shown). Mutations that increased the activity of FtsWI or RodA-PBP2 are colored orange, whereas mutations that inhibit the activity of these SEDS-bPBP protein complex are colored magenta. The FtsW putative catalytic residue is colored red. Most of the active/inactive mutations are located at the interaction interface between ECL4 of *Tt*RodA and the tip of the lower part of the pedestal domain of *Tt*PBP2.

(4), the main function of FtsN is to activate the essential septal PG synthase FtsWI. FtsN does this by initiating two parallel signaling pathways, one in the cytoplasm and one in the periplasm (Fig 1B). In the cytoplasm, the cyto domain of FtsN interacts with FtsA to switch it to an active conformation which contributes to FtsWI activation by an unknown mechanism [1,18,27]. In the periplasm, the E domain of FtsN acts through FtsQLB, likely causing a conformational change such that the AWI domain of FtsL becomes available to interact with FtsI which stimulates the activity of FtsW [1,18,19,22,27]. This proposed activation pathway is strikingly similar to that of the analogous RodA-PBP2 complex despite the regulatory proteins being completely unrelated. In this complex MreC interacts with the pedestal domain of PBP2 and is proposed to switch the complex to the active conformation [15,16,38,39]. However, a detailed mechanism by which these SEDS-bPBPs are stimulated by their activators is still not clear. The *ftsW* and *ftsI* mutations isolated and characterized here that affect their activation *in vivo* are confined to the interface of the pedestal domain of FtsI and ECL4 of FtsW. Our results are consistent with the proposed activation mechanism for FtsWI [16,21] and provide new insights into the working mechanism of the SEDS-bPBP complexes.

Previous studies showed that mutations in the CCD domain of FtsB or FtsL enable cells to divide with reduced or no FtsN function [18–20], whereas mutations in the AWI domain of FtsL (L86F, E87K), which lie on the opposite side of a putative helix encompassing the CCD domain, abolish FtsQLB's ability to activate septal PG synthesis *in vivo* and to stimulate FtsWI *in vitro* [21,22]. However, these inactive FtsL mutations are suppressed by a mutation in FtsW (M269I) which was isolated as a suppressor of an ATPase mutant of FtsEX and found to

tolerate the depletion of FtsN [20,22]. Here we show that this mutation contains a unique substitution that allows the bypass of FtsN when overexpressed indicating input from FtsN is no longer required. However, drastic substitutions at the M269 position (M269K/E) result in inactive FtsW proteins that are key for teasing out the signaling cascade for FtsWI activation.

Two additional mutations were isolated that suppress these inactive FtsL mutations, one in the ECL4 of FtsW (E289G) and one in the pedestal domain of FtsI (K211I). Notably, the *ftsW^{E289G}* mutation was also isolated as a suppressor of *ftsN* deletion by other groups [4]. Both of these mutants bypass the essentiality of FtsN indicating that they are active mutants. They also suppress the ATPase mutant of FtsEX that blocks the cytoplasmic signal from FtsN. Strikingly, cells with a single copy of *ftsW^{E289G}* allows cells to divide almost normally without FtsN. FtsW^{E289G} can also suppress FtsI mutants (G57D, S61F and R210C) defective in activation *in vivo*. However, none of the reported active mutants of the upstream proteins can suppress an FtsW mutant (M269K) severely defective in activation, although they are able to suppress one (A270T) that is weakly defective. These results suggest that FtsW^{M269K} is locked in an inactive state and are in agreement with the model in which the periplasmic signaling cascade for FtsWI activation follows the order: ^{E}FtsN>FtsQLB (FtsL^{AWI})>FtsI>FtsW [22]. Mutations in upstream proteins of the signaling cascade can block the activation of FtsWI, but once FtsW is in the active conformation (induced by the active mutations E289G or M269I reported here), this FtsN-triggered activation signaling cascade can be bypassed.

This signaling cascade (^{E}FtsN>FtsQLB [FtsL^{AWI}]>FtsI>FtsW) can explain why FtsI homologs in several Gram-positive organisms are essential even though their transpeptidase activity is not [40,41]. In *Bacillus subtilis*, cells continue to divide in the presence of a β-lactam that blocks the transpeptidase activity of the FtsI homolog (PBP2B) and a catalytically inactive mutant of PBP2B supported cell division [40]. Under these circumstances, the transpeptidase activity required for crosslinking PG glycan strands synthesized by FtsW is likely provided by another nonessential PBP (PBP3), which localizes to the division site and becomes essential only when the transpeptidase activity of PBP2B is blocked [40]. Comparable results were obtained in *Staphylococcus aureus*, in which introduction of a point mutation in the transpeptidase active site of the FtsI homolog (PBP1) did not result in a division block but the gene could not be deleted [41]. We suggest that the essential role of the FtsI homologs in these organisms is to transmit the activation signal from FtsL^{AWI} to FtsW to start polymerization of PG glycan strands. Thus, cells can still divide even when FtsI homolog is inhibited by a β-lactam because FtsW is still activated and another PBP crosslinks the newly synthesized PG glycan strands. However, in *E. coli*, which lacks an accessory PBP, inhibition of the transpeptidase activity of FtsI results in the futile cycle of septal PG synthesis by FtsW [42].

How FtsI stimulates the activity of FtsW is not currently clear. Previous studies on the RodA-PBP2 complex indicated that the interaction between the pedestal domain of bPBP and the ECL4 of their cognate SEDS protein plays a critical role in activating the polymerization activity of the SEDS proteins [16]. In support of this, mutations in the ECL4 of RodA or in the pedestal domain of PBP2 activate the complex [15,17]. Similarly, here we found that mutations in the ECL4 of FtsW can lock the protein in an inactive state (M269K) that no longer responds to the activation by FtsN or in an active state (E289G and to a less extent M269I) that no longer requires FtsN for division. Mutations in the pedestal domain of FtsI can also activate (K211I) or inactivate (G57D, S61F, L62P and R210C) the FtsWI complex *in vivo*. Such mutations are quite unique and are unlikely to be isolated by site-directed mutagenesis. In BTH assays they do not detectably affect the interaction of FtsW with FtsI or their interaction with other divisome components. However, it is highly likely that these mutations, which are clustered to a small region at the interaction interface between the pedestal domain of FtsI and ECL4 of FtsW, lock the complex in an active or inactive state (Fig 10). This change will be difficult to

detect using classical genetic or biochemical approaches because previous work on the RodA-PBP2 complex indicates that this interaction interface lacks specific interactions and is transient and flexible [16]. Nonetheless, the isolation of active mutations in both FtsWI and RodA-PBP2 at the same positions suggest that they are activated in similar ways by completely unrelated proteins, which likely modulate the interface between the SEDS protein and the bPBP.

Biochemical characterization of the FtsW and FtsI mutations studied here will be key to understanding their effect on the FtsWI complex, but our genetic analysis provides some clues. Analysis of the substitutions at the 269 position of FtsW suggests that a hydrophobic interaction at this position is critical for FtsWI to transition between the active and inactive conformations. A strong hydrophobic interaction will favor the protein in the active form, whereas loss of this hydrophobic interaction will favor the inactive form. On the other hand, the negative charge of E289 does not seem to be important for FtsW function as even changing it to arginine only slightly affects its function. However, only the E289G substitution can lock FtsW in the active form and surprisingly, it is epistatic to the M269K mutation that locks the protein in the inactive form. Similarly, the positive charge of K211 of FtsI is not essential for its function, but only K211I produces an active FtsI protein and suppresses nearby inactive FtsI mutations. These observations suggest that subtle local re-arrangements of the FtsWI complex, likely involving the region around M269-A270 and E289 of FtsW and region around K211 of FtsI, are critical for the stabilization of the active and inactive conformations. Since these mutations appear to lock FtsWI in the active/inactive conformation, they should be useful for structural and biochemical analysis of the FtsWI complex to reveal details of the activation mechanism.

## Materials and methods

### Media, bacterial strains, plasmids and growth conditions

Cells were grown in LB medium (1% tryptone, 0.5% yeast extract, 0.5% NaCl and 0.05 g/L thymine), LB medium without NaCl or 1% NaCl at indicated temperatures. When needed, antibiotics were used at the following concentrations: ampicillin = 100 μg/ml; spectinomycin = 25 μg/ml; kanamycin = 25 μg/ml; tetracycline = 25 μg/ml; and chloramphenicol = 20 μg/ml. Strains, plasmids and primers used in this study are listed in S1 Table, S2 Table and S3 Table, respectively. Construction of strains and plasmids is described in detail in S1 Text with the primers listed in S3 Table.

### BTH assay

To detect the interaction between FtsW/FtsI or its mutants and other division proteins, appropriate plasmid pairs were co-transformed into BTH101. The next day, single colonies were resuspended in 1 ml LB medium, and 3 μL of each aliquot was spotted on LB plates containing ampicillin, kanamycin, 40 μg/ml X-gal and 100 μM IPTG. Plates were incubated at 30°C overnight before imaging.

### Localization of GFP-fusion proteins

*Localization of FtsW-L60-GFP*. Overnight cultures of SD237 [W3110, leu::Tn10 *ftsW*::*kan*/ pDSW406 (P$_{BAD}$::*ftsW*, *cat*)] carrying plasmid pDSW210 (P$_{206}$::gfp) or pSD247 (P$_{206}$::*ftsW-l60-gfp*) and its derivatives were diluted 1:100 in fresh LB medium with antibiotics and 0.2% arabinose, grown at 30°C for 2 h. Cells were then collected by centrifugation and washed twice with fresh LB and resuspended in the same volume of LB medium. These arabinose-free

culture were then diluted 1:10 in fresh LB medium and IPTG was added to a final concentration of 60 μM. 4 hours post removal of arabinose and induction with IPTG, cells were immobilized on 2% agarose pad for photograph.

*Localization of GFP-FtsI.* Overnight cultures of SD288 [W3110, *ftsW::kan gfp-ftsI/pSD257 (pSC101ts, ftsW, Spc)]* carrying plasmid pBAD33 or pDSW406 (pBAD33, P$_{BAD}$::ftsW) and its derivatives from 30˚C were diluted 100X in fresh LB medium with antibiotics, grown at 30˚C for 2 h. The cultures were then diluted 1:10 in fresh LB medium with antibiotics, 0.2% arabinose and 10 μM IPTG and grown at 37˚C for 3 hours. Cells were immobilized on 2% agarose pad for photograph.

## Creation of FtsW and FtsI mutant library and screen for suppressors of dominant-negative FtsL mutants

*Creation of FtsW and FtsI mutant libraries.* *ftsW* was subjected to random PCR mutagenesis using the primer 5-NcoI-ftsW and 3-ftsW-HindIII and plasmid pSEB429 (pDSW208, P$_{204}$::*ftsW*). Gel purified PCR product was digested with NcoI and HindIII and ligated into pSEB429 digested with the same enzymes. Ligation product was transformed into JS238 competent cells and plated on LB plates with ampicillin and glucose. About 20,000 transformants were pooled together and plasmids were isolated and stocked. FtsI mutant library was constructed similarly using primer pair pLY91-I-F and pLY91-I-R and plasmid pLY91 as a template.

*Screen for suppressors of dominant-negative FtsL mutants.* The mutagenized pSEB429 (pSEB429M) or pLY91 (pLY91M) was transformed into strain SD399 [W3110, *ftsL::kan /* pSD256 (pSC101ts, *ftsL, aadA*)] harboring plasmid pSD296-E87K (pBAD33, P$_{BAD}$::*ftsL$^{E87K}$*) and suppressors of FtsL$^{E87K}$ were selected on LB plates with antibiotics, 0.2% arabinose and 60 μM IPTG at 37˚C. Strain SD399 could not grow at 37˚C because plasmid pSD256 was not able to replicate, plasmid pSD296-E87K was not able to complement as FtsL$^{E87K}$ was not functional and toxic when induced with arabinose. Transformants that could grow on the selective plates thus likely contained plasmids expressing FtsW or FtsI mutants that could suppress the defect of FtsL$^{E87K}$. In the screen with FtsW mutant library, 20 transformants were picked and restreaked on the selective plates for growth, 14 of them grew well. Plasmids were isolated from these 14 transformants and retransformed into strain SD399 carrying pSD296-E87K to confirm the suppression. Sequencing of the *ftsW* genes in these 14 suppressing plasmids showed that all of them harbored the E289G mutation (8 contained only the E289G mutation, 6 contained the E289G mutation as well as additional mutations). However, these other mutations were not further studied because the suppression of FtsL$^{E87K}$ was apparently due to the E289G mutation. In the screen with FtsI mutant library, 52 transformants were picked and restreaked on the selective plates for growth, 18 of them grew well. 17 of the plasmids isolated were confirmed to suppress FtsL$^{E87K}$ and sequencing of the *ftsI* gene on these plasmids showed that all of them harbor the K211I mutation.

## Western blot

To measure the level of FtsW-L60-GFP mutants, overnight cultures of JS238 (wild type strain) harboring the respective plasmid were diluted 1:100 in LB medium with ampicillin and 30μM IPTG. After growth at 37˚C for 2 hours, OD$_{600}$ of each culture was measured and samples were taken for western blot. Cells were collected, resuspended in SDS-PAGE sample buffer and kept at room temperature for 30 min before they were loaded on the SDS-PAGE gel for analysis. Western blotting and detection of FtsW-L60-GFP were performed as previously described for other division proteins [43]. Anti-GFP antibody (Rockland Immunochemicals) was used at a dilution of 1/1000.

## Supporting information

**S1 Text. Construction of strains and plasmids.**
(DOCX)

**S1 Table. Bacterial strains used in this study.**
(DOCX)

**S2 Table. Plasmids used in this study.**
(DOCX)

**S3 Table. Primers used in this study.**
(DOCX)

**S1 Fig. Chromosomal FtsW$^{M269I}$ tolerates depletion of FtsN but cannot bypass FtsN.** (A) Spot test of the ability of FtsW$^{M269I}$ to tolerate the depletion of FtsN. Colonies of strain SD264 [W3110, *ftsN::kan*/pBL154 (pSC101$^{ts}$, P$_{syn}$::*ftsN*, *spc*)] and SD265 [W3110, *ftsW$^{M269I}$ ftsN::kan*/pBL154 (pSC101$^{ts}$, P$_{syn}$::*ftsN*, *spc*)] was resuspended in 1 ml of LB medium, serially diluted. 3μl of each dilution was spot on LB plates with antibiotics. Plates were incubated at 30˚C for 24 hours or at 42˚C overnight and photographed. (B) Morphology of strain SD264 and SD265 at 30 and 42˚C. Overnight culture of SD264 and SD265 were diluted 1:100 in fresh LB medium with antibiotics and grown at 30˚C for 2 hours. Samples were taken for photograph or further diluted in 1:10 in LB medium and shifted 42˚C. The cultures were kept in exponential phase and samples were taken for photograph at indicated time points after the temperature shift. (C) P1 transduction to test the ability of FtsW$^{M269I}$ to bypass FtsN. P1 transduction of *ftsN::kan* from strain CH34/pMG20 (*ftsN::kan*) was attempted into S3 (W3110 *leu::Tn10*) and SD247 (W3110 *leu::Tn10, ftsW$^{M269I}$*) following a standard procedure. Transductants were selected on LB plates with kanamycin and 1 mM sodium citrate. Plates were incubated at 30, 37, and 42˚C overnight or up to 24 hours. Obtained transdutants were restreaked on LB medium with kanamycin and 1 mM sodium citrate for further growth. Only the results from 42˚C was shown. Scale bar is 5 μm.
(TIF)

**S2 Fig. P1 transduction to test the ability of FtsW$^{M269I}$ to bypass FtsN.** P1 transduction of *ftsN::kan* from strain CH34/pMG20 (*ftsN::kan*) was attempted into W3110 and SD247 (W3110 *leu::Tn10, ftsW$^{M269I}$*) harboring plasmid pSEB417 (P$_{204}$::*ftsN*), pSEB429 (P$_{204}$::*ftsW*) or pSEB429-M269 (P$_{204}$::*ftsW$^{M269I}$*) following a standard procedure. Transductants were selected on LB plates with ampicillin, kanamycin, 1 mM sodium citrate and with or without 60 μM IPTG. Plates were incubated at 30˚C up to 24 hours or 37 or 42˚C overnight. Similar number of transductants were obtained for a control expressing FtsN and when FtsW$^{M269I}$ was induced with 60μM IPTG. Only the results from 37˚C was shown.
(TIF)

**S3 Fig. Alignment of the extracellular loop 4 (ECL4) of FtsW and RodA from diverse bacterial species.** Amino acid sequences were obtained from NCBI, aligned with Clustal Omega and then created using ESPRIPT: http://esprit.ibcp.fr/. FtsW: *E. coli* (gi|2132970), *K. pneumoniae* (gi|597728208), S. flexneri (gi|110613701), *Y. pestis* (gi|115346361), *B. thailandensis* (gi|685745844), S. enterica (gi|205337406), *P. aeruginosa* (gi|15599609), *L. pneumophila* (gi|295650162), *M. xanthus* (gi|108761950), *C. crescentus* (gi|426019958), *A. tumefaciens* (gi|586950133), *B. fragilis* (gi|598888368), *B. subtilis* (gi|2493592), *L. monocytogenes* (gi|424013134), *E. faecalis* (gi|323480530), *S. aureus* (gi|384230086), *M. tuberculosis* (gi|613782430), *C. glutamicum* (gi|674168391), *B. burgdoferi* (gi|2493585), *T. thermophiles*

(BAW00664.1). RodA: *E. coli* (gi|78101784), *S. flexneri* (gi|78101785), *K. pneumonia* (gi|499531772), *S. enterica* (gi|16501890), *P. aeruginosa* (gi|15599197), *L. pneumophila* (gi|148280678), B. *thailandensis* (gi|83652485), *V. cholera* (gi|126519168), *Y. pestis* (gi|115348295), *C. crescentus* (gi|220963725), *B. subtilis* (gi|732351), *M. xanthus* (gi|108462975), *L. monocytogenes* (gi|336024336), *S. pneumoniae* (gi|302638595), *M. tuberculosis* (gi|444893486), *C. glutamicum* (gi|62388939), *T. thermophilus* (WP_011228544.1).
(TIF)

**S4 Fig.** Bacterial two hybrid test of the interaction between FtsW (A) or FtsI (B) and other divisome proteins. Plasmids pairs were transformed into strain BTH101, the next day a single transformant of each resulting strain was resupended in 1 ml LB medium, 3µl of each culture was spot on LB plates containing antibiotics, 40 µg/ml X-gal and IPTG. Plates were incubated at 30˚C for about 24 hours before photographing.
(TIF)

**S5 Fig. Complementation test of FtsL mutants.** Plasmid pBAD33, or pSD296 ($P_{BAD}$::*ftsL*) or its derivatives with different *ftsL* allele were transformed into strain SD399 (W3110, *ftsL*::*kan* /pSD256) harboring plasmid and transformants selected on LB plates with antibiotics and glucose. The next day, a single colony of each resulting strain was resuspended in 1 ml LB and serially diluted by 10. 3 µl of each dilution was spot on LB plates with appropriate antibiotics, with or without arabinose. Plates were incubated at 30˚C for 24 hours or at 37˚C overnight and photographed.
(TIF)

**S6 Fig.** FtsW$^{E289G}$ (A) and FtsI$^{K211I}$ (B) provide resistance to the division inhibitory activity of FtsE$^{D162N}$X. plasmid pSD221 (pEXT22, $P_{tac}$::*ftsEX*) or pSD221-D162N (pEXT22, $P_{tac}$::*ftsE$^{D162N}$X*) were transformed into strain W3110, SD247 (W3110, *leu*::*Tn10*, *ftsW$^{M269I}$*), SD488 (W3110, *leu*::*Tn10*, *ftsW$^{E289G}$*) or LYA8 (W3110, *leu*::*Tn10*, *ftsI$^{K211I}$*). The next day, a single transformant of the resulting strains was resuspended in 1 ml BL medium and serially diluted in 10. 3 µl of each dilution was spotted on LB plates with antibiotics and with or with IPTG. Plates were incubated at 37˚C overnight and photographed. Note that the resistance of SD488 to FtsE$^{D162N}$X was stronger than that of strain SD247.
(TIF)

**S7 Fig.** Overexpression of FtsI$^{K211I}$ (A) or FtsW$^{E289G}$ (B) tolerates the depletion of FtsN. Plasmid pDSW208, pLY91 (pDSW208, $P_{204}$::*ftsI*), pSEB429 (pDSW208, $P_{204}$::*ftsW*), or their derivatives carrying different *ftsI* or *ftsW* allele were transformed into strain SD264 [W3110, *ftsN*::*kan* /pBL154 (pSC101$^{ts}$, *ftsN*)] and transformants selected on LB plates with antibiotics and glucose at 30˚C. The test was performed as Fig 3B.
(TIF)

**S8 Fig. FtsI$^{K211I}$ and FtsW$^{E289G}$ bypass FtsN.** (A) P1 transduction to test the ability of FtsI$^{K211I}$ (A) and FtsW$^{E289G}$ (B) to bypass FtsN. P1 transduction of *ftsN*::*kan* from strain CH34/pMG20 (*ftsN*::*kan*) was attempted into W3110, LYA8/pLY105 [W3110, *leu*::*Tn10*, *ftsI$^{K211I}$* / $P_{204}$::*ftsI$^{K211I}$*] and SD488 (W3110, *leu*::*Tn10*, *ftsW$^{E289G}$*) following a standard procedure. In the case of LYA8/pLY105, transductants were selected on LB plates with 1% NaCl, kanamycin, 1 mM sodium citrate and different concentration of IPTG at 30, 37, and 42˚C. Transductants were only obtained at 42˚C and they can only grow at 42˚C when restreaked. For SD488, transductants were selected on LB plates with kanamycin and 1 mM sodium citrate at 30, 37, and 42˚C. A similar number of transductants were obtained at all three temperatures and only the result from 37˚C is shown. 4 transductants from SD488 (W3110, *leu*::*Tn10*,

*ftsW*$^{E289G}$) were restreaked on the same selection plates at 37˚C and all grew well.
(TIF)

**S9 Fig. FtsW**$^{E289G}$ **is a unique mutant that suppresses the depletion of FtsN.** (A) Complementation test of the FtsW mutants using strain SD237. (B) Spot test of the ability of FtsW mutants to rescue the growth of FtsN depletion strain at non-permissive condition. The tests were done as in Fig 3.
(TIF)

**S10 Fig. FtsI**$^{K211I}$ **is a unique mutant that suppresses the depletion of FtsN.** (A) Complementation test of the FtsI mutants. Plasmids pDSW208, pLY91 (P$_{204}$::*ftsI*) and pLY105 (P$_{204}$::*ftsI*$^{K211I}$) were transformed into strain PS413 (W3110, *ftsI23*). A single transformant of each resultant strain was resuspended in 1 ml LB and serially diluted by 10. 3 μl of each dilution was spot on LB plates with antibiotics and with or without IPTG. Plates were incubated at 30˚C for 24 hours or at 42˚C overnight and photographed. (B) Spot test of the ability of FtsI mutants to rescue the growth of FtsN depletion strain at non-permissive condition. Test was done as in S7B Fig.
(TIF)

**S11 Fig. Overexpression of FtsN and FtsA**$^{R286W}$ **suppress FtsW**$^{A270T}$ **but not FtsW**$^{M269K}$. (A) spot test of the ability of FtsN overexpression to suppress FtsW$^{A270T}$ and FtsW$^{M269K}$. (B) Spot test of the ability of FtsA$^{R286W}$ to suppress FtsW$^{A270T}$ and FtsW$^{M269K}$. The test was carried out as in Fig 9 but in strain SD295 [W3110, *ftsW::kan recA56 slrD::Tn10* /pSD257 (pSC101$^{ts}$, *ftsW*)] or SD390 [W3110, *ftsA*$^{R286W}$, *ftsW::kan recA56 slrD::Tn10* /pSD257(pSC101$^{ts}$, *ftsW*)]. Note that FtsW$^{A270T}$ was unable to complement the FtsW depletion strain but FtsA$^{R286W}$ suppressed the defect completely. In addition, in the presence of FtsA$^{R286W}$, less inducer was needed to induce wild type FtsW for complementation.
(TIF)

## Acknowledgments

We thank members of the Du lab and the Lutkenhaus lab for comments and advice in preparing the manuscript.

## Author Contributions

**Conceptualization:** Joe Lutkenhaus, Shishen Du.

**Data curation:** Ying Li, Shishen Du.

**Formal analysis:** Ying Li, Joe Lutkenhaus, Shishen Du.

**Funding acquisition:** Joe Lutkenhaus, Shishen Du.

**Investigation:** Ying Li, Han Gong, Rui Zhan, Shushan Ouyang, Kyung-Tae Park.

**Methodology:** Ying Li.

**Project administration:** Joe Lutkenhaus, Shishen Du.

**Resources:** Joe Lutkenhaus.

**Supervision:** Joe Lutkenhaus, Shishen Du.

**Visualization:** Ying Li.

**Writing – original draft:** Ying Li, Shishen Du.

**Writing – review & editing:** Ying Li, Joe Lutkenhaus, Shishen Du.

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
