## [Decision Letter · Decision Letter 0]

22 Feb 2021

Dear Dr Du,

Thank you very much for submitting your Research Article entitled 'Genetic analysis of the septal peptidoglycan synthase FtsWI complex supports a conserved activation mechanism for SEDS-bPBP complexes' to PLOS Genetics.

The manuscript was fully evaluated at the editorial level and by independent peer reviewers. The reviewers appreciated the attention to an important topic but identified some concerns that we ask you address in a revised manuscript

We therefore ask you to modify the manuscript according to the review recommendations. Your revisions should address the specific points made by each reviewer.

[LINK]

Yours sincerely,

Daniel B. Kearns

Associate Editor

PLOS Genetics

Lotte Søgaard-Andersen

Section Editor: Prokaryotic Genetics

PLOS Genetics

Shishen,

Three reviewers (and myself) were very enthusiastic about your manuscript. We found it timely, important, and well-conducted. Nonetheless, the reviewers provide good comments for you to consider as part of a revision. Nice work, and thanks for submitting to PLoS Genetics.

Dan

Reviewer's Responses to Questions

**Comments to the Authors:**

Reviewer #1: This manuscript by Li et al provides significant further genetic evidence and molecular details supporting a model of interaction/ activation of SEDS family glycosyltransferase FtsW by its class b PBP partner FtsI during E. coli cell division. They further characterize a previously isolated activation mutant of FtsW (M269I) and isolate additional mutants of FtsW (E289G) and FtsI (K211I) that further clarify a main pathway of activation of E. coli divisome peptidoglycan synthesis (FtsN  FtsL (in association with FtsQB)  FtsI  FtsW), with FtsA and FtsEX also figuring into this regulated septal cell wall synthesis. The authors do an excellent job of using genetics to tease out interactions between all of these players and a hierarchy of their activities, building/confirming existing models proposed for the SEDS/bPBP partner interactions in cell length growth (RodA/PBP2). Consideration of their results when mapping mutations onto RodA/PBP2 structure interactions likely analgous to FtsW/FtsI reveal that activation of FtsW through FtsI occurs via the extracellular loop 4 (ECL4) of FtsW with the 'pedestal domain' of FtsI, separate from regions responsible for recruiting these proteins to the divisome. At the same time, their results demonstrate that the specific residues within these domains that are most integral for activation show some variation between species.

Overall I believe the manuscript is solid, with conclusions well-supported by the data. I only have some minor comments:

- The initial results with the A270T mutation of FtsW are interesting and also become important for later portions of the paper in that it serves as a partially 'inactive' form. This is stated within lines 172 - 173, with reference to figure SF4, but seems a foundational piece of data that bears more stressing and explanation here. I would actually recommend that SF4 be made a regular figure given the importance of A270T later and that its 'defect' is otherwise not apparent to the reader.

- The caveat to lines 174 and 175 would be that the gfp adds to stability compared to a non-fused version of the mutants. The 'inactivity' of these even with the gfp though would be consistent with their statements/arguments. I would just clarify.

- The cells carrying ftsN::kan in Figure 2 panel B seem to be mostly lysing (pock-marked). Does the necessity of antibiotics here make this more prevalent perhaps? Doesn't seem to fully 'suppress' PG defects then?

- Has an M269L allele been attempted, would they predict that might act like the Ile version?

- Line 259. All substitutions affect protein structure by definition. Just add something like 'substantially'.

- Lines 262 - 264, it was unclear to me initially why one would necessarily predict this exactly. This may be in part due to my still not grasping at this point what the 'defect' with A270T was.

- In Figure 4 the GFP-FtsW and GFP-FtsI signal in the M269K background seems to be infrequent and particularly irregular. Is this really still able to localize to the divisome, or are we seeing residual preformed rings that never disassembled as FtsW was depleted (maybe still containing some residual W?) The GFP-FtsW label in panel A should also be put in.

- The text needs editing in terms of putting in commas. Several sentences are awkward or confusing without commas properly separating clauses, etc.

Reviewer #2: This paper from Li and co-workers investigates the activation of cell wall synthesis within the divisome by the FtsWI synthase complex. The authors use an elegant series of genetic experiments to show that amino acid substitutions in the pedestal domain of FtsI and extracellular loop 4 of FtsW can either lead to hyperactivation of the synthase or its inactivation. Epistasis studies provided additional support for the pathway of activation within the divisome triggered by FtsN flowing from FtsQLB to FtsI to FtsW. Mapping of the activating and inactivating changes in FtsWI to the structure of the related SEDS-bPBP cell wall synthase RodA-PBP2 from the elongation machinery provides evidence for a conserved activation mechanism between the two synthases as changes that activate RodA-PBP2 map to the same region of the structure. Overall, this is a very nice paper that makes an important contribution to our understanding of how cell wall synthesis is activated within morphogenic machines. I have only minor points to be addressed.

1) Line 67: evidences are > evidence is

2) Line 90: ...AWI domain of FtsL is LIKELY TO BE exposed...

3) Line 244-245: Thus, these two mutations do not... I think this statement is too strong since it is based solely on two-hybrid analysis. Please moderate it.

4) Line 271: ...the weak mutant... change to ...the weakly defective mutant...

5) Line 273: Similarly, it would be better to describe the other mutant as the "severely defective mutant".

6) Line 287: Change "inactive mutations" to "inactivating mutations".

7) Line 307: ...whether THEY are...

8) Table 1: The text is not colored as indicated in the legend.

9) Line 336: ...resulting in... is too strong. Change to "likely causing".

10) Line 410: I think you mean POSITIVE charge of K211

11) Some of the spot assay figures could be better labeled to make it easier for the reader to follow. For example, Fig. 5A, it is not clear from looking at the figure that FtsL is depleted by a temperature shift. Please review and modify figures accordingly to make them easier to follow.

Reviewer #3: 1. Could the authors clarify the relationship between two possible activation pathways, one form FtsN, and one from FtsEX, on the activation of FtsW? Are they independent and redundant of each other? FtsW superfission mutants M269I and E289G can suppress FtsE(D162N)X or FtsN depletion separately. Can they suppress both at the same time (FtsE(D162N)X + FtsN depletion)?

2. I am not sure about the statement that the transpeptidase activity of FtsI is not essential, Antibiotics such as Aztreonam specifically inhibits FtsI’s transpeptidase activity leads to the cease of cell division, long filamentation, and eventually death. I think I understand the authors’ point of view that another essential function of FtsI is to activate FtsW, for which FtsI’s transpeptidase activity is not required. This point should be clarified.

3. The authors’ observations that the negative charges of E289 and K211 are not essential are interesting. They imply that the interactions between FtsW and FtsI may not be typical electrostatic protein-protein interactions. Could the authors offer more specific insight regarding how this could be possible based on the structural model, other than “subtle local re-arrangements of the FtsWI complex”? for example, some type of structural model?

4. I am not sure about the epistatic analysis of FtsW-E289G on M269K and A270T. To me it is an effect of E289G overwriting M269K and A270T, but this overwriting function is not dependent on the presence of M269K or A270T. In other words, the super activation phenotype of W289G is not dependent on M269K or A270T, as it can compensate FtsW depletion. M269K reduced the activity of E289G, certainly indicating some type of interactions between the two sites but does not mean that one mutant is dependent on the other to function. Or am I understand the concept of epistasis in the wrong way?

Minor comments:

1. Lines 110-124: Including clarification in the text to Figs. S1A and S1B will allow readers to better match the text and figure for a better understanding.

2. Lines 276-296: Including clarification in the text to Figs. 9A and 9B will allow readers to better match the text and figure for a better understanding.

3. Page 17: In the Table 1 description it lists “Active (green)” however there is no green text in the table or text.

4. Figure 10 (structure with mutations) and Table 1 (mutations) are related, but minor changes will make the connection stronger. Figure 10 has increased activity mutations in orange, inhibiting mutations in magenta, and the catalytic residue in red. Table 1 has active mutations in green. Keeping a consistent color scheme between the structure figures and the table would be helpful.

**Have all data underlying the figures and results presented in the manuscript been provided?**

Reviewer #1: Yes

Reviewer #2: Yes

Reviewer #3: Yes

PLOS authors have the option to publish the peer review history of their article (what does this mean?). If published, this will include your full peer review and any attached files.

Reviewer #1: No

Reviewer #2: No

Reviewer #3: No

---

## [Editor Report · Decision Letter 1]

18 Mar 2021

Dear Dr Du,

We are pleased to inform you that your manuscript entitled "Genetic analysis of the septal peptidoglycan synthase FtsWI complex supports a conserved activation mechanism for SEDS-bPBP complexes" has been editorially accepted for publication in PLOS Genetics. Congratulations!

Yours sincerely,

Daniel B. Kearns

Associate Editor

PLOS Genetics

Lotte Søgaard-Andersen

Section Editor: Prokaryotic Genetics

PLOS Genetics

Comments from the reviewers (if applicable):

**Data Deposition**

http://datadryad.org/submit?journalID=pgenetics&manu=PGENETICS-D-21-00068R1

**Press Queries**

---

## [Editor Report · Acceptance letter]

9 Apr 2021

PGENETICS-D-21-00068R1 

Genetic analysis of the septal peptidoglycan synthase FtsWI complex supports a conserved activation mechanism for SEDS-bPBP complexes 

Dear Dr Du, 

We are pleased to inform you that your manuscript entitled "Genetic analysis of the septal peptidoglycan synthase FtsWI complex supports a conserved activation mechanism for SEDS-bPBP complexes" has been formally accepted for publication in PLOS Genetics! Your manuscript is now with our production department and you will be notified of the publication date in due course.

With kind regards,

Alice Ellingham

PLOS Genetics

On behalf of:
